# Drying in the low-latitude Atlantic Ocean contributed to terrestrial water storage depletion across Eurasia

Zexi Shen[1,2], Qiang Zhang [1,2✉], Vijay P. Singh[3,4], Yadu Pokhrel [5], Jianping Li [6,7], Chong-Yu Xu [8] & Wenhuan Wu[1,2]

Eurasia, home to ~70% of global population, is characterized by (semi-)arid climate. Water scarcity in the mid-latitude Eurasia (MLE) has been exacerbated by a consistent decline in terrestrial water storage (TWS), attributed primarily to human activities. However, the atmospheric mechanisms behind such TWS decline remain unclear. Here, we investigate teleconnections between drying in low-latitude North Atlantic Ocean (LNATO) and TWS depletions across MLE. We elucidate mechanistic linkages and detect high correlations between decreased TWS in MLE and the decreased precipitation-minus-evapotranspiration (PME) in LNATO. TWS in MLE declines by ~257% during 2003-2017 due to northeastward propagation of PME deficit following two distinct seasonal landfalling routes during January-May and June-January. The same mechanism reduces TWS during 2031-2050 by ~107% and ~447% under scenarios SSP245 and SSP585, respectively. Our findings highlight the risk of increased future water scarcity across MLE caused by large-scale climatic drivers, compounding the impacts of human activities.

[1] State Key Laboratory of Earth Surface Processes and Resource Ecology, Beijing Normal University, 100875 Beijing, China. [2] Faculty of Geographical Science, Beijing Normal University, 100875 Beijing, China. [3] Department of Biological and Agricultural Engineering and Zachry Department of Civil & Environmental Engineering, Texas A&M University, College Station, TX, USA. [4] National Water and Energy Institute, UAE University, Al Ain, UAE. [5] Department of Civil and Environmental Engineering, Michigan State University, East Lansing, MI, USA. [6] Frontiers Science Center for Deep Ocean Multispheres and Earth System/Key Laboratory of Physical Oceanography/Academy of the Future Ocean, Ocean University of China, 266100 Qingdao, China. [7] Laboratory for Ocean Dynamics and Climate, Pilot Qingdao National Laboratory for Marine Science and Technology, 266237 Qingdao, China. [8] Department of Geosciences and Hydrology, University of Oslo, Oslo, Norway. ✉email: zhangq68@bnu.edu.cn

**G**lobal land area and population under extreme-to-exceptional terrestrial water storage (TWS) drought would increase by 4% and 5%, respectively, by the end of the twenty-first century owing to substantial decline in TWS across many regions[1]. Such declines in TWS have major implications for water sustainability and food security. Eurasia, home to ~70% of the global population[2], is one of such critical global regions where securing sufficient water storage is crucial to maintain future water supplies and food security[3–6]. However, declines in water storage in the past decades have been seriously threatening freshwater availability, especially over the mid-latitude (N30°-N60°) portion of Eurasia[7–9]. Observations from the Gravity Recovery and Climate Experiment (GRACE) satellite mission indicate significant and consistent decline in TWS across mid-latitude Eurasia (MLE) during 2002–2017;[9–11] TWS is the sum of river water, groundwater, soil moisture, canopy water and snow (ice) water[12–14]. Moreover, TWS is almost the sole freshwater resource in arid regions, supporting domestic, industrial, and agricultural water needs. Thus, the declining TWS across Eurasia is directly threatening socio-economic growth and sustainable development in the region[15,16]. This calls for a critical need to better understand the causes behind consistently decreasing TWS over MLE.

Anthropogenic activities have been suggested to be among the key factors causing TWS loss[9,17–24]. Decreased TWS in the Northwest China (Xinjiang), the North China Plain, the Middle East and the Southwest Russia during 2002–2017 was attributed to agricultural irrigation, mining dewatering and/or domestic water withdrawals[25–29]. By contrast, it is surprising to find synchronously and consistently declining TWS across Eurasia[9], implying that these TWS changes could be driven mainly by climate forcing[1]. These previous findings imply that TWS changes in Eurasia could be driven by both climatic variations and human activities. However, a mechanistic understanding of the exact causes of TWS changes across Eurasia is lacking.

Atmospheric teleconnection analyses have been widely used in quantifying relations between regional water cycle changes and large-scale meteorological circulations from the perspective of contemporary correlation[30–32]. The El Niño-Southern Oscillation (ENSO), Arctic Oscillation and North Atlantic Oscillation were viewed as the dominant factors influencing TWS loss across Asia and eastern Europe[11,33], while further study is indispensable from the viewpoint of water vapor transport[34], highlighting meteorological mechanisms behind the teleconnections and source-sink relations of water vapor[35]. Here, we used the Lagrangian transport and dispersion model to depict the large-scale atmospheric moisture transport, shedding new light on the links between the regional water cycle and the large-scale atmospheric patterns across Eurasia[36–39].

Atmospheric rivers[35,40,41], as water vapor transport channels, convey moisture variation-induced impacts from source to target regions[40]. Moisture deficits over oceans were demonstrated to migrate landward as precipitation-minus-evaporation (PME) deficits (meteorological droughts) triggering landfalling meteorological droughts over 16% of global continents[42,43]. However, there is a gap in our understanding of teleconnections between moisture deficit over oceans and TWS shortage over the continents from a viewpoint of water vapor transport. The North Atlantic Ocean (NATO) was identified as the major source of water vapor for Eurasia[44,45]. Thus, the key scientific rationale behind this study is that the large-scale ocean-land moisture transport by the westerlies could be the atmospheric causes behind the TWS shortage across Eurasia. The central science question the study addresses is: what are various mechanisms and pathways whereby climatic changes over the ocean impact or drive TWS variations across Eurasia, and how much of the

attribution in seasonal sense. Answer to this question can enhance the mechanistic understanding of water resource variability and availability at the continental scale and in a changing climate.

Here, to investigate into this moisture transport mechanism, we use the Lagrangian transport and dispersion model to simulate backward moisture transport across Eurasia, taking Xinjiang in China (Fig. 1) as the water vapor sink region (Method 1), where the Tarim basin and the Junggar basin channel water vapor from Europe to Asia (Fig. 1)[46,47]. Meanwhile, normalization is performed for the trend items of PME in both land and ocean, TWS anomaly, sea surface temperature (SST), and total contribution rates (TCR) of water vapor, respectively (Method 2). We use the Pearson correlation and cross-correlation methods to evaluate teleconnections between PME over NATO and the PME and TWS across MLE (Methods 3–4). Further, we use the modified Mann-Kendall trend detection method to evaluate the significance of trends in PME over oceans and in TWS over continents (Method 5). We find that the PME across MLE is co-influenced by the decline in PME over NATO1-3 (sub-regions 1-3 of NATO, Fig. 2) and increase in PME over NATO4 (sub-region 4 of NATO, Fig. 2), which likely explain the low correlation between the two. Different from the PME in lands, the TWS decline across the MLE is highly coherent with the predominant decline in PME over NATO1-3. Moreover, we develop random forest models based on the ECMWF Reanalysis v5 (ERA5) data and GRACE RL05 data for the period 2003–2017 to simulate teleconnections between PME within NATO1 and NATO3 (Fig. 2) and TWS within sub-regions across MLE (Fig. 2). The random forest models are then used to derive TWS changes during 2018–2050 based on the multi-model ensemble data from the Coupled Model Intercomparison Project Phase 6 (CMIP6) under the Shared Socio-economic Pathway 2-4.5 (SSP245) and SSP585 scenarios (Method 7). All abbreviation and detailed clarification are listed in the Supplementary Table 1.

## Results
**Linkage between water vapor over NATO and TWS across MLE.** The Tarim and Junggar basins in Xinjiang, China, channel water vapor from Europe to Asia (Fig. 1)[46,47]. Thus, we proposed Xinjiang as the sink of water vapor and simulated water vapor transport across Eurasia during 2003–2017 using the Lagrangian transport and dispersion model called the FLEXPART (Fig. 1, Method 1). All source regions of water vapor for Xinjiang, detected by the FLEXPART model, mainly included Asia (AS, excluding Xinjiang), Europe (EU, including Russia), North Atlantic Ocean, and other ten source regions of water vapor (Fig. 1 and S1, 2). We defined the total contribution rate (TCR) as the fractional ratio equaling the sum of water vapor from source regions excluding loss and/or input of water vapor along the routes divided by all released water vapor in Xinjiang (Method 1). The total water vapor input from the above regions forms the precipitation in Xinjiang. The simulated TCR was verified and validated based on the ERA5 and GPCC (Global Precipitation Climatology Centre) precipitation in Xinjiang.

Of all potential source regions of water vapor, Xinjiang has the long-term averaged contribution rate (CR) of only ~0.1% (Supplementary Fig. 2), corroborating that water vapor changes over Xinjiang can be attributed mainly to water vapor variations in the outer source regions. The long-term averaged CR for AS, EU and NATO were 30.1%, 11.0%, and 2.8%, respectively (Fig. 1 and S2). The cumulative CRs (AS, EU and NATO) of >80% occurred over 86.7% of the entire study period (Fig. 1c), showing that water vapor entering Xinjiang was mainly from AS, EU, and NATO. However, even though NATO is one of the important

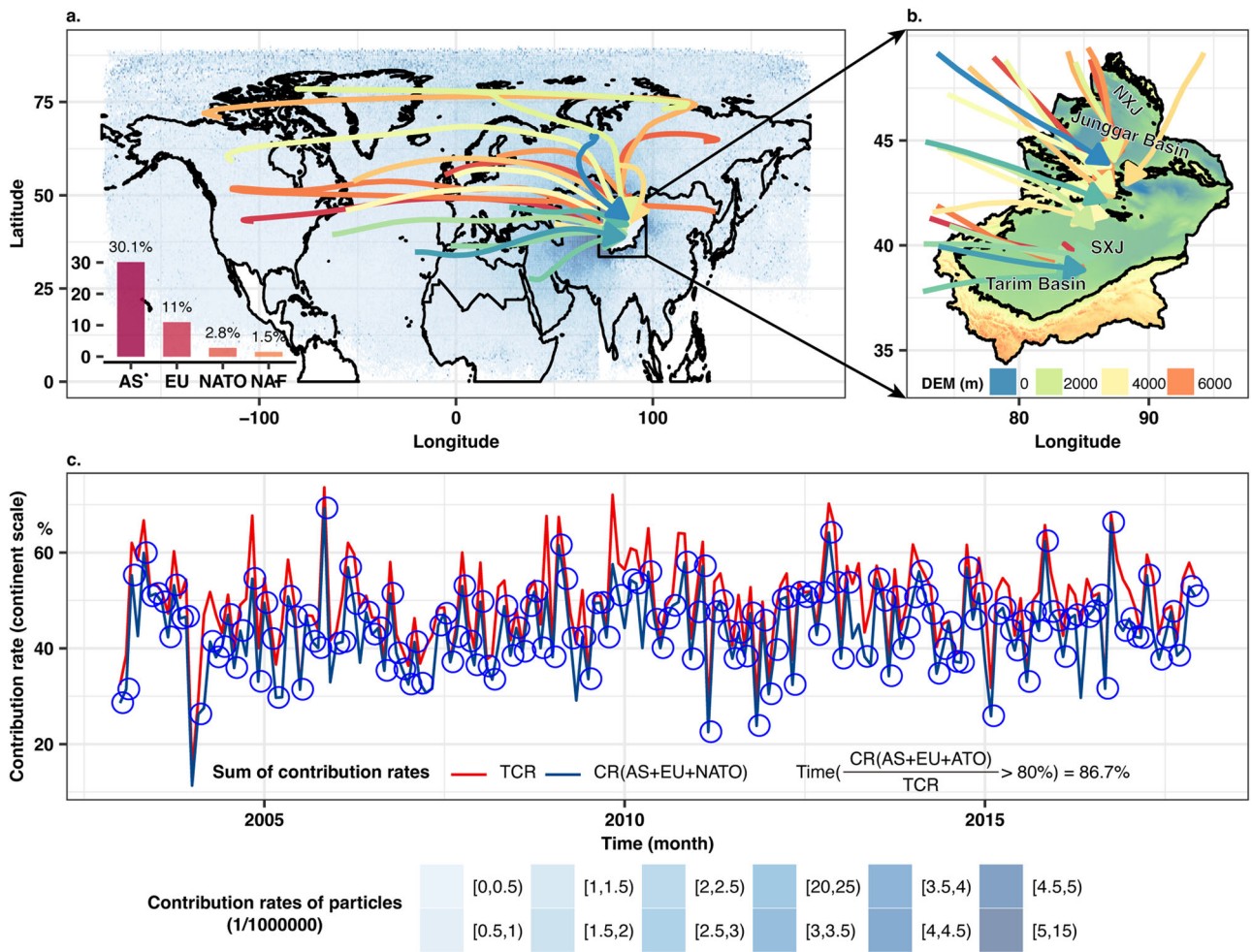

**Fig. 1 Identification of the water vapor sources for Xinjiang, China.** Contribution rates (CRs) of 500,000 water particles to the precipitation in Xinjiang during 2003–2017 by month, and cluster analysis of back traced trajectories of water particles (**a**). The routes of moisture trajectories and spatial pattern of terrain in Xinjiang (**b**). Temporal covariation of the total contribution rate (TCR) and the sum of CRs from Asia (AS), Europe (EU) and the North Atlantic Ocean (NATO) (CR(AS + EU + NATO)) during 2003–2017 (**c**). The time when CR(AS + EU + NATO)/TCR > 80% took account of 86.7% of the whole study period. The NXJ and SXJ refer to the north and south Xinjiang in the plot. The NAF in the subplot a refers to North Africa.

water vapor sources for AS and EU[44,45], due to water vapor loss during the water vapor transport, the CR of NATO was smaller than the CRs of AS and EU. Further analyses categorized AS, EU, and NATO into AS1-2, EU1-3 and NATO1-4, respectively, based on locations of water vapor particles (Supplementary Fig. 3). High correlation between PMEs in Xinjiang of China and AS1-2 verified the channel function of Xinjiang in water vapor transport from Europe to Asia (Supplementary Figs. 3–5).

However, against the intermediate role of Europe in the moisture transport across Eurasia, we detected low and negative correlations (−0.43 to 0.12) between PMEs in EU1-3 and PMEs in both southern and northern Xinjiang, respectively (Supplementary Fig. 3d). This could be attributed to inverse changing tendencies of PMEs in NATO1-3(declining) and NATO4 (increasing) during 2003–2017, implying the complexity of mixed sources of PMEs in EU1-3 (Supplementary Figs. 3 and 4). Moreover, the Tian Mountain lying across central Xinjiang obstructs southward movement of water vapor from high-latitude NATO2 and NATO4, primarily influencing PME in northern Xinjiang and hence low correlations between PMEs in NATO1-3 and PME in northern Xinjiang (Supplementary Figs. 3 and S5). By contrast, PME in southern Xinjiang with water vapor mainly from NATO1-3 was highly correlated with PME in NATO1-3

instead of PME in NATO4 (Supplementary Figs. 3 and 4). When compared to PME, TWS could be influenced by the long-term impacts of climatic variations in water vapor source regions. The TWS in EU1 and AS1-2, being influenced mainly by PME in NATO1-3, decreased during 2003–2017, and was highly correlated to the TWS in southern and northern Xinjiang (correlation coefficients ranged between 0.51 and 0.93) (Supplementary Figs. 6–8). Based on these findings and water vapor transport across Eurasia (Fig. 1), we identified an eastward and landward water vapor transport route, i.e., EU1-AS1-XJ-AS2.

**Meteorological drying in LNATO-depleted TWS across MLE.** Aforementioned analyses indicate that the PME across the Eurasia is co-influenced by declining PMEs over NATO1-3 and increasing PME over NATO4, involving two inversed trends in continental PME, which further impairs the capability of continental PME in reflecting TWS variations. In contrast, the TWS across Eurasia reflects the predominant trends in PME over ocean by the westerly induced atmospheric connection. Moreover, the PME deficits over NATO1-3, especially in NATO3, could exacerbate TWS loss in EU1, AS1, XJ and AS2. To further understand the spatiotemporal pattern of TWS across MLE under the impact of climate change over NATO1-4, we subdivided MLE

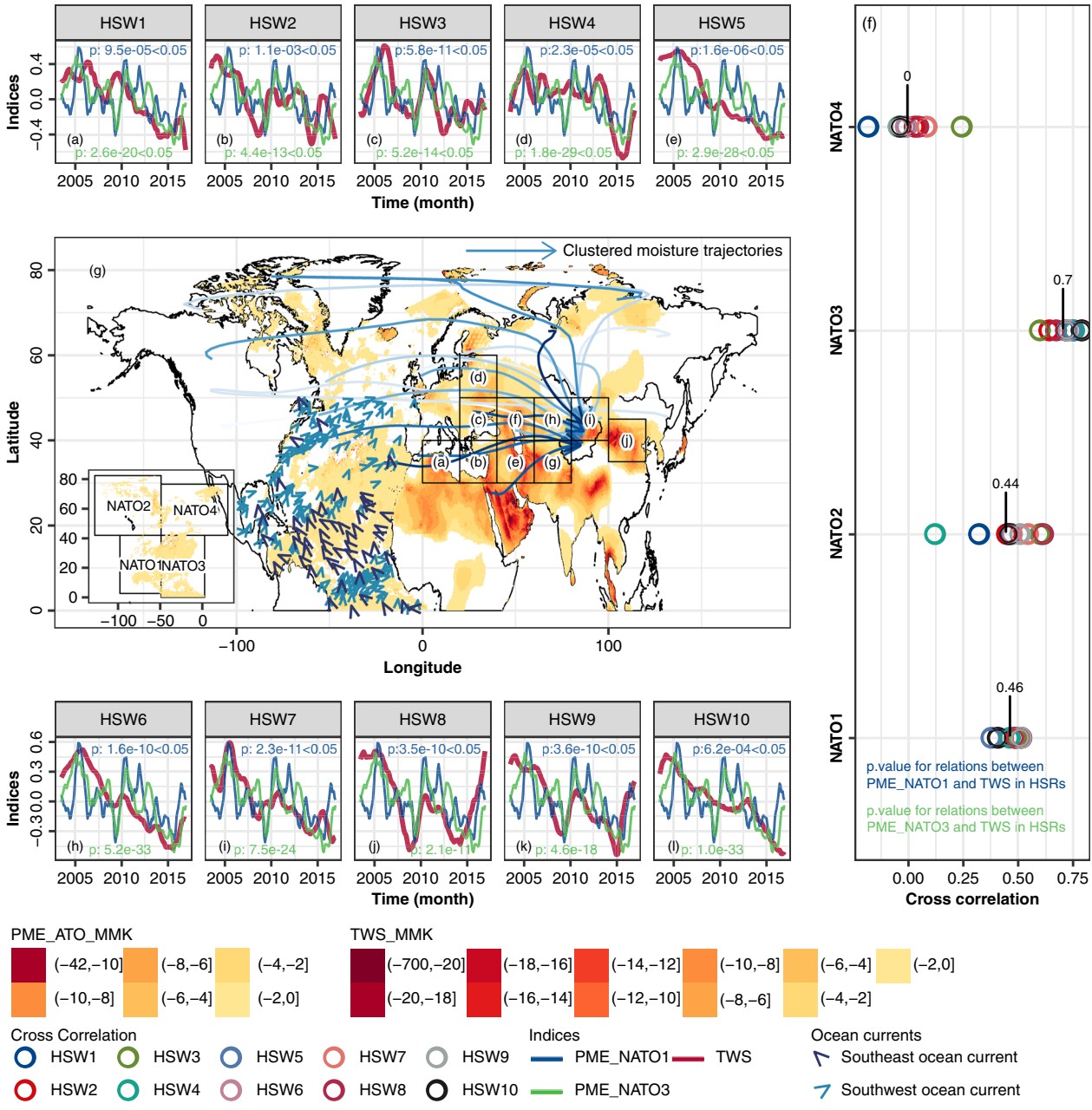

**Fig. 2 Spatiotemporal evolutions in PME over NATO and TWS across Eurasia.** Temporal covariations among the normalized trend items of the regional sums of terrestrial water storages (TWSs) in sub-regions 1–10 (HSRs 1–10) across mid-latitude (N30°–N60°) Eurasia and the normalized trend items of the regional sums precipitation-minus-evapotranspiration (PMEs) in NATO1 and NATO3 (**a–e** and **h–l**, respectively). Spatial patterns of the modified Mann-Kendall (MMK) trends of PMEs in the NATO and TWSs in North Africa, Europe and Asia (**g**). Cross Pearson's correlation between PMEs in northern NATOs and TWSs in HSRs 1–10 (**f**). PME_NATO_MMK refer to the MMK trend in PMEs over NATO. TWS_MMK refers to the MMK trend in TWS across Eurasia and North Africa. The ocean currents in the plot were the long-term averaged ocean currents in NATO based on the OSCAR (Ocean Surface Current Analysis Real-time) dataset during 1993–2019. The moisture trajectories in subplot (**g**) were clustered using the *k*-means cluster method based on the locations of particles in the trajectories.

into 19 sub-regions ($10° \times 10°$). Then we further defined 10 sub-regions where TWSs were highly correlated with PME in NATO3 (the cross-correlation > 0.50) as high-correlated sub-regions (HSR), and defined other sub-regions as low-correlated sub-regions (LSR) (Supplementary Figs. 9 and 10).

Spatially, HSR1-10 formed a successive channel of water vapor transport, transmitting the impact of PME variations over NATO along MLE to TWS in Xinjiang (Supplementary Fig. 11). The

PME deficits over LNATO caused the TWS loss along HSR1-10 via water vapor transport by the northward ocean currents and subsequent landward landfalling of PME deficits by the westerlies during 2003–2017 (Fig. 2g). In this case, we detected the average cross correlations of 0.46 and 0.7 between steadily decrease in PME over LNATO (NATO1 and NATO3) and decrease in TWS along HSR1-10 (Fig. 2 and S9). On the contrary, the PME variations over high-latitude NATO (NATO2 and NATO4) and

the decrease of TWS along the HSR1-10 were less consistent, even uncorrelated, in both space and time with the averaged cross correlations of 0.44 and 0 (Fig. 2 and S9).

The SST variations could be one of the driving factors causing PME deficits over NATO[48,49]. Here, we evaluated the relation between SST and PME during 2003–2017 using the ERA5 reanalysis historical data. We found that declining SST within N0°–N20° NATO (the modified Mann-Kendall (MMK) trends ranged between (−4, 0]) and slightly increasing SST within N20°–N40° NATO (MMK trends ranged between (0, 2]) resulted in the decreased PME (MMK trends ranged between (−2, 0]) (Supplementary Fig. 12a, b, c, e, g). The drastic warming SST (MMK trends ranged between (2, 8]) tended to increase PME over NATO1. In general, the SST variations were positively correlated with the PME changes over low-latitude (N0°–N40°) NATO. This finding is consistent with previous studies[48,49]. Besides, we found reversed relations between SST and PME over high-latitude NATO when compared to LNATO (Supplementary Fig. 12h). The drastic cooling SST (MMK trends ranged between (−6, 0]) steadily increased PME over N40°–N80° NATO (Supplementary Figs. 12a, b, f, h).

**Seasonal landfalling routes of PME deficits from NATO to MLE.** We successfully identified two major seasonal landfalling routes of the PME deficit over NATO. During January to May, the PME deficit occurred within the N0°–N30° NATO, away from the northern westerly belt of N30°–N60°. Then, the eastward landfall of PME deficit influenced the adjacent North Africa, and finally reached the Northern China Plain (NCP) across the Caspian Sea (CS) and Xinjiang by the westerlies (Fig. 3 and S13a–e), which formed the first landfalling routes of PME deficit over NATO. During January to September, we found the northward propagation of PME deficit from N0°–N30° NATO to N30°–N60° NATO. However, due to the time-lag effect of drying over N30°–N60° NATO, the eastward landfalling of PME deficit from N30°–N60° NATO started in June and lasted up to January of the subsequent year (Fig. 3a, b). This was another seasonal landfalling route of the PME deficit from N30°–N60° NATO, and then the propagation of PME deficit continued due to the northern westerlies from Europe to NCP via CS and Xinjiang. Intercomparison of the above-mentioned two seasonal landfalling routes of PME deficits helped distinguish two overlapping joints, i.e., the CS and Xinjiang. Therefore, TWSs in the CS and Xinjiang, influenced by PME changes over NATO, were both in serious PME deficit at the annual scale (Fig. 3a, c and Supplementary Fig. 13). Meanwhile, results from the maximum covariance analysis showed that the regions with synchronous variations between TWS and PME over the low-latitude NATO matched well with the two seasonal landfalling routes in the first leading mode with an explained variance of 28.72%, which further evidenced that these landfall routes of PME deficit were sourced from the NATO (Supplementary Fig. 14 and Supplementary Text 1).

**Projected TWS variations across MLE during 2018–2050.** We depicted the projected spatiotemporal patterns of TWS over HSR1-10 and multi-model ensembled CMIP6 PMEs over NATO1 and NATO3 under the SSP245 and SSP285 scenarios during 2018–2050 (Fig. 4). The PMEs over NATO1 and NATO3 and TWSs across the whole HSR1-10 slightly increased during the period of 2018–2031 under the SSP245 and SSP585 scenarios (Fig. 4a, b). Since the variation in TWS across HSR1-10 was mainly impacted by the PME variation over NATO3 in comparison with the PME variation over NATO1 (Supplementary Fig. 15), the intensification in PME deficit over NATO3 would accordingly enhance the TWS loss across HSR1-10. Thus, during

2031–2050, the scenario change from SSP245 to SSP585 intensified the depletion of PME over NATO3 from 77% to 686% and the depletion of TWS across the whole HSR1-10 from ~107% to ~447% (Fig. 4a, b). Besides, given variable performance of CMIP6 models in simulating the PME over the NATO, we further projected the TWS over HSR1-10 based on the weighted multi-model ensembled CMIP6 PMEs over NATO1 and NATO3 (Supplementary Text 2). During 2018–2050, the originally projected TWS and the bias-corrected TWS revealed similar TWS changes across Eurasia under SSP585 (Supplementary Fig. 17b). However, under SSP245, the originally projected TWS increased by ~111% during 2018–2031 while the bias-corrected TWS decreased by ~291% during 2018–2031, and then decreased by ~107% and ~447% while bias-corrected TWS decreased by ~28% and ~473% during 2031–2050 under SSP245 and SSP585 scenarios, respectively (Supplementary Fig. 17a).

The spatial patterns of MMK trends in PMEs over NATO indicated that the PME deficits over LNATO were transmitted by the northward ocean currents to mid-latitude Europe under the SSP245 and SSP585 scenarios during 2018–2050, forming a distinct route of PME deficits originating from LNATO (Fig. 4c, d). Then, PME deficits were transmitted by the westerlies across Eurasia (Figs. 3 and 4). Furthermore, the PME deficit over LNATO was intensified and the impact of PME deficit propagating in TWS along Eurasia was amplified when the scenario transferred from SSP245 to SSP585 scenario during 2018–2050 (Fig. 4c, d).

Previous studies mainly attributed widespread declines in TWS across the Eurasia to the anthropogenic water withdrawals without considering the consistency among the declines in TWS across the Eurasia[9]. An attribution analysis indicated that TWS changes over 71% of the regions—where the total freshwater withdrawals (TFW) had negative impacts on TWS variations—were influenced mainly by PME variations over NATO, whereas TWS changes over 29% of these regions were impacted primarily by TFW (Supplementary Fig. 18). Besides, an evaluation of interactions between TFW and TWS suggested that variations in TWS across the MLE can affect water withdrawals at the regional scale, influencing regional economic development (Supplementary Fig. 19 and Supplementary Text 3). Moreover, from a mechanistic perspective, we evidenced that the drying over the LNATO in the past decades contributed to the consistent and synchronous TWS depletion across the MLE. We also demonstrated that the same mechanism would drive TWS declines in the future. Meanwhile, we identified two seasonal landfalling routes of meteorological drying from LNATO: (1) LNATO-North Africa-Caspian Sea-Xinjiang-NCP during January-May and (2) LNAO-North-Africa-Europe-Caspian Sea-Xinjiang-NCP during June–January.

In sum, we demonstrated the mechanistic linkages between meteorological drying in LNATO (oceanic meteorological behaviors) and decline in TWS across MLE (continental water availability), clarifying and elucidating the key atmospheric drivers behind the decline in TWS across the MLE. These findings indicate that atmospheric drivers have played key role in the persistent and broader TWS declines across MLE. The findings of this study have important implications for water resources management and food security at continental scale in a changing climate.

## Methods

**Lagrangian diagnostic of *E − P* for Xinjiang, China.** The Lagrangian particle dispersion model FLEXPART was initially developed for forward and backward simulation of the air mass and water vapor transport driven by the observed or reanalysis meteorological datasets, quantitatively detecting the source regions of water vapor for a specific target region[38]. From the perspective of particles, the model dispersed the air mass as plenty of water particles, and simulated the movement and variation of specific humidity of every particle in the observed or reanalysis atmospheric environment per 6 h. During each simulation time step, for each individual particle, the

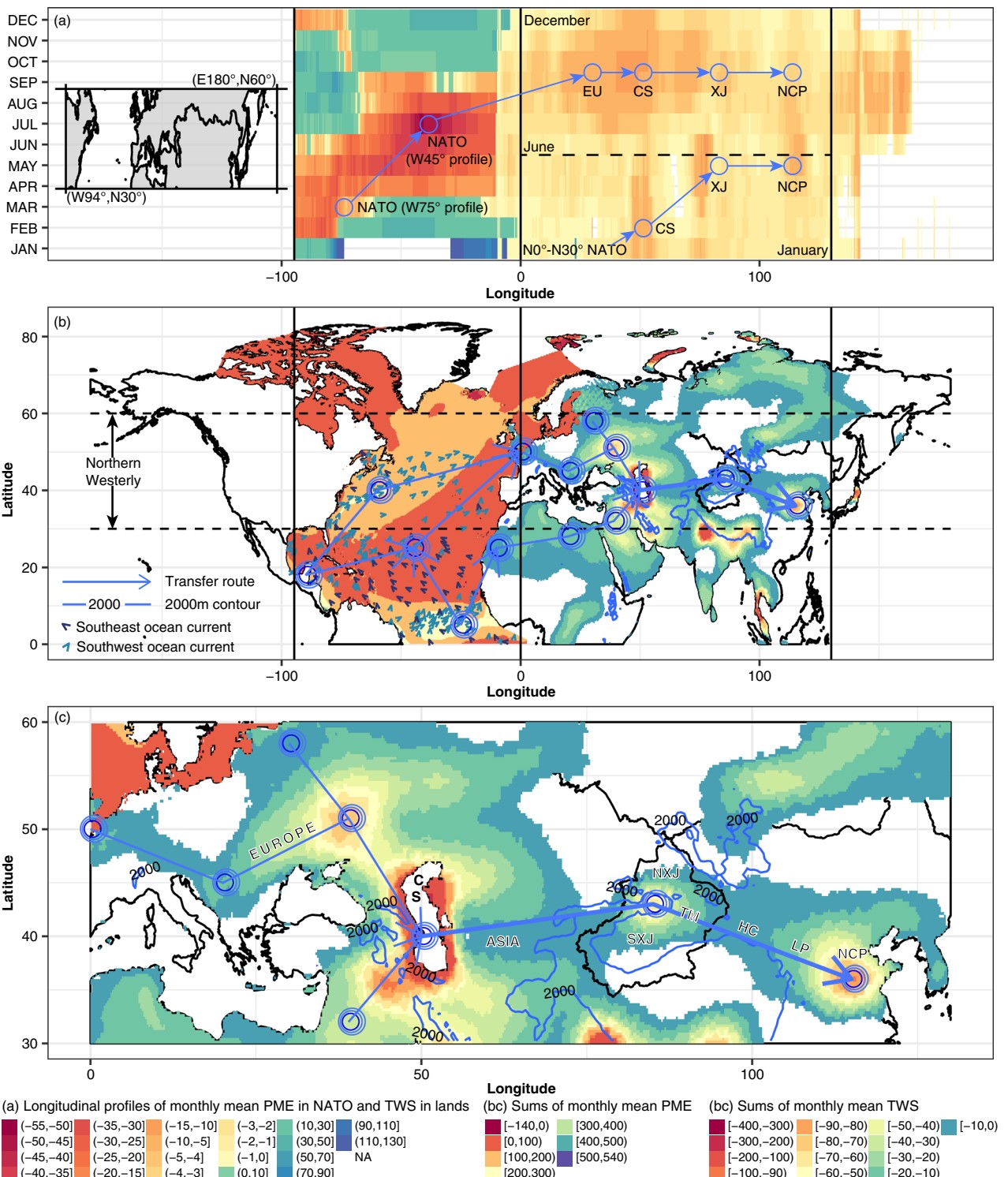

**Fig. 3 Seasonal landfalling paths of PME deficit from NATO.** Longitudinal profile of monthly averaged mean precipitation-minus-evapotranspiration (PME) anomalies in the mid-latitude (N30°–N60°) Atlantic Ocean (NATO) and the monthly mean terrestrial water storage (TWSs) anomalies in mid-latitude (N30°–N60°) North Africa (NAF) and Eurasia from January to December (**a**). Spatial patterns of the sum of monthly mean PME anomalies (unit: mm) in NATO and the sum of negative monthly mean TWS anomalies (unit: mm) in lands during 2003–2017 (**b**). Transfer routes of the landfalling impacts of deficits in PMEs over northern NATO from ocean to land during 2003–2017 (**c**). The NXJ, SXJ, TM, HC, LP and NCP in subplot c refer to North Xinjiang, South Xinjiang, Tian Mountain, Hexi Corridor, Loess Plateau and North Chain Plain, respectively.

balance ($e - p$) between evaporation ($e$) and precipitation ($p$) was calculated as the variation in the specific humidity along the trajectory of the particle as follows:

$$e - p = m\frac{dq}{dt} \tag{1}$$

where $e$ and $p$ are the variation rates of evaporation and precipitation in a particle over time, respectively; $q$ is the specific humidity, and $m$ is the mass of the particle[38]. The $e - p$ refers to the variation rate of water vapor content in a particle over time. When $e - p < 0$, the particle loses the water vapor content in terms of precipitation; when $e - p > 0$, the particle recharges the water vapor content in terms of evaporation.

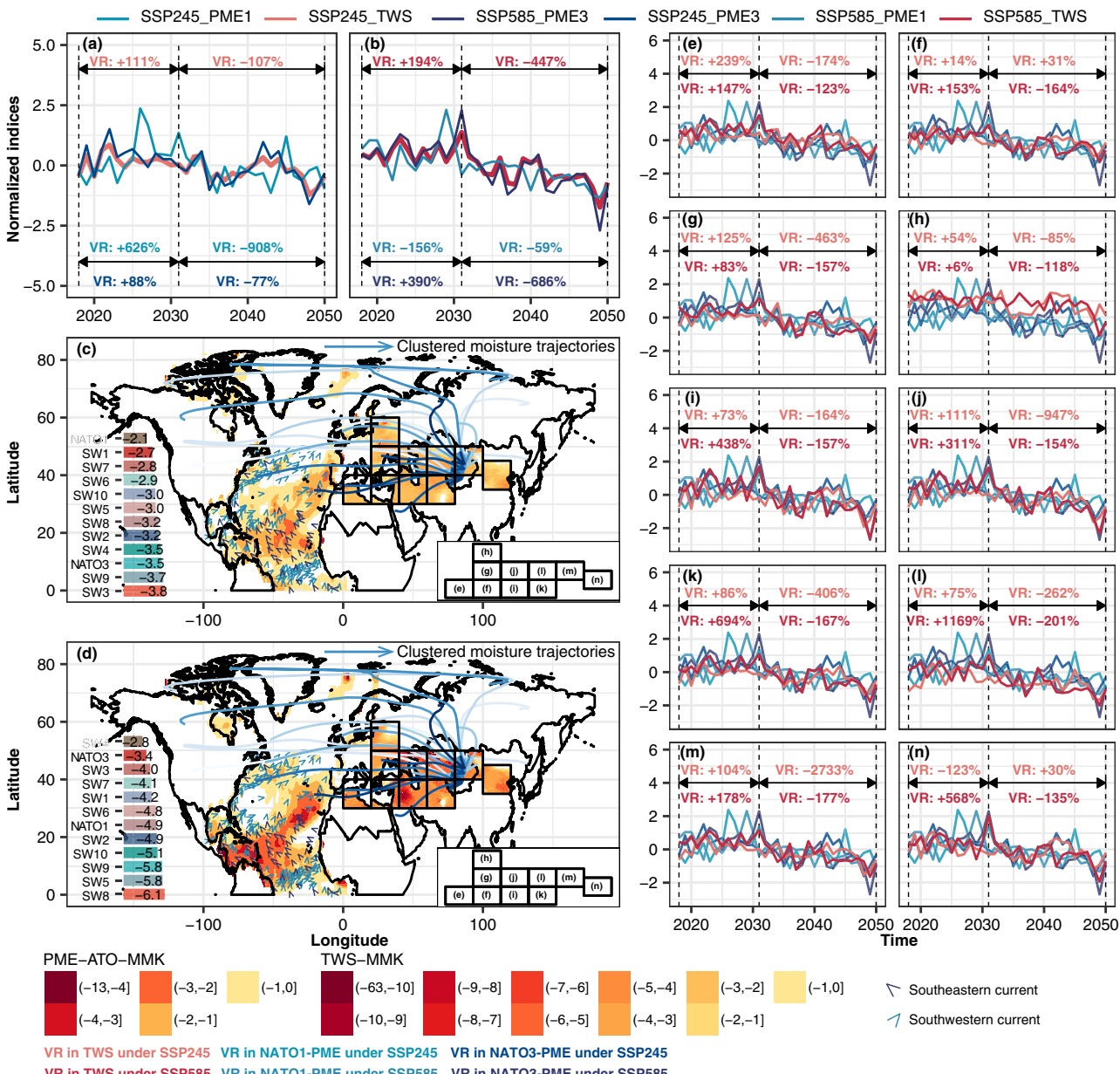

**Fig. 4 Projected spatiotemporal evolutions in PME over NATO and TWS in HSR1-10.** Temporal variations in the annual-sum normalized trend items of the zonal sums of TWSs across the whole HSR1-10 and variations in the annual-sum normalized trend items of zonal sums of PMEs in NATO1 and NATO3 under SSP245 and SSP585 during 2018–2050 (**a**, **b**). Spatial patterns of the modified Mann-Kendall (MMKs) trends of PME over NATO and TWS in HSR1-10 under SSP245 (**c**) and SSP585 (**d**) during 2018–2050 (**c**, **d**). Temporal variations in the annual-sum normalized trend item of the zonal sum of TWS over every HSR and annual-sum normalized trend items of the zonal sums of PMEs in NATO1 and NATO3 under SSP245 and SSP585 during 2018–2050, respectively (**e–n**). The VR in the plot refers to the varying rates of indices during the time interval, i.e., 2018–2031 and 2031–2050. The moisture trajectories in subplot (**c**, **d**) were categorized using the *k*-means cluster method based on the locations of water particles.

For the target area, the cumulative balance ($E - P$) between evaporation ($E$) and precipitation ($P$) is calculated as the sum of the $e$-$p$ in all particles in the areas over time:

$$E - P = \frac{\sum\limits_{n=1}^{n}(e-p)}{A} \quad (2)$$

where $E - P$ and $e - p$ refer the variation rates of water vapor contents in the target area and in the specific particle, respectively; $A$ is the area of the target region; and $n$ is the number of all particles in the target area.

This study backward simulated 500,000 particles in Xinjiang using FLEXPARTv10.0 at 6-hourly time step based on the ERA-interim meteorological data (0.75° × 0.75°) during 2003–2017. Before the simulation, the ERA-interim data were downloaded and preprocessed using flex_extract software (https://www.flexpart.eu/downloads), which contained reanalysis wind filed at 61 vertical levels from 0.1 to 1000 hPa[50]. During the study period, we performed 180

backward simulations from day 1 to day 10 at 6-hourly time step for every month. Each simulation result was used to approximately reflect the water vapor transport for each month. All trajectories extracted from the simulation results were clustered as 20 trajectories during 2003–2017 using the *k*-means cluster method[50]. Further, to reflect the monthly patterns of variations in trajectories, based on the monthly mean of particle locations in 2003–2017, monthly clustered trajectories had been calculated using the *k*-means cluster method as well[51].

Since the water vapor content in a particle was recharged or released along the trajectory, the variation in the transport route could not represent the contribution of water vapor from the source region to the target region[52]. Thus, we first classified the potential source regions by the continent and ocean boundaries, which included Asia (AS, excluding Xinjiang), Europe (containing Russia, EU), North Africa (NAF), North America (NAM), North Atlantic Ocean (NATO), Caspian Sea (CS), Mediterranean Sea (MS), Arctic Ocean (AO), Black Sea (BS), Indian Ocean (IO), Red Sea (RS), Xinjiang (XJ), and Pacific Ocean (PO). To filter

out particles sourced from the regions (except AS, XJ, and Russia without transport route) that lost all of the water vapor contents recharged from the source regions before crossing the boundary of Xinjiang, the selected particles finally released in Xinjiang should meet the following condition (Eqs. 3–5).

$$(E-P)_{source} = \frac{\sum_{s=1}^{s}(e-p)}{A_{source}} \qquad (3)$$

$$(E-P)_{route} = \frac{\sum_{r=1}^{r}(e-p)}{A_{route}} \qquad (4)$$

$$(E-P)_{source} - abs((E-P)_{route}) > 0 \qquad (5)$$

where $(E-P)_{source}$ referred to total variation in the water vapor contents in particles, which were in a source region, and $abs((E-P)_{route})$ referred to the absolute total variation in the water vapor contents in particles, which were in the route from a source region to the target region (XJ). The $s$ and $r$ were the number of particles in a source region and in the route from a source region to XJ, respectively. As for particles sourced from the AS (excluding Xinjiang) and Russia, there was no inter route to XJ, and particles sourced from these regions should meet the demand that $(e-p)_{source} > 0$. For the particles sourced from XJ itself, we concerned how much the water vapor content released in XJ. Thus, $(e-p)_{source}$ in XJ should meet the demand that $(e-p)_{source} < 0$.

After the selection of particles meeting the above demands, the contribution rates of water vapor contents in particles from the source region to XJ were calculated as follows:

$$num_{select} = \sum_{k=1}^{k} num_k \qquad (6)$$

$$(E-P)_k = \frac{\sum_{1}^{num_k}(e-p)}{A_{XJ}}(e-p) < 0 \qquad (7)$$

$$(E-P)_{all} = \frac{\sum_{1}^{n}(e-p)}{A_{XJ}}(e-p) < 0 \qquad (8)$$

$$CR_k = \frac{(E-P)_k}{(E-P)_{all}} \times 100\% \qquad (9)$$

where $num_{select}$ refers to the number of all selected particles released in Xinjiang; $num_k$ refers to the number of the selected particles from the $k$th ($k = 1, 2, 3…13$) source region; $(E-P)_k$ refers to the total release of the water vapor contents in selected particles from the kth source region; $(E-P)_{all}$ refers to the total release of the water vapor contents in all particles in Xinjiang; and $CR_k$ refers to the contribution rates of water vapor contents in particles from the $k$th source region.

Even though the FLEXPART model could simulate the water vapor movement from the perspective of particles, since it was mainly driven primarily by reanalysis data, the accuracy of simulation results highly depended on the quality of data. Hence, to verify the simulation results of FLEXPRT and avoid the impact of periodicities in time series indices, we performed a temporal covariation analysis and Pearson's correlation analysis between the normalized TCR from all source regions ($\sum_{1}^{k} CR_k$) and the normalized PMEs based on GPCC (Global Precipitation Climatology Centre) and ERA5 (ECMWF Reanalysis 5th generation)-based PMEs (Supplementary Table 2). The GPCC and ERA5 precipitation datasets have been previously evaluated with in-situ observations, demonstrating that these two datasets accurately represented observed precipitation variations[53].

**Calculation of normalized indices.** To compare the covariations in meteorological indices and terrestrial water storage (TWS) indices, and avoid the impact of periodicities in time series, this study calculated the normalized trend items of indices for further analysis as Eq. 10. The trend item of the time series index was extracted using the moving average of the index.

$$Normalized\ Index = \frac{trend_{item} - mean(trend_{item})}{max(trend_{item}) - min(trend_{item})} \qquad (10)$$

where $trend_{item}$ refers to the trend item of every index in the study. Since the original index for the terrestrial water storage (TWS) was an anomaly, the normalized trend item of TWS was calculated as Eq. 11.

$$Normalized\ TWS = \frac{TWS_{anomaly}}{max(TWS_{anomaly}) - min(TWS_{anomaly})} \qquad (11)$$

In this study, abbreviations of all indices referred to the normalized trend items of the zonal sums of indices if there was no specific denotation.

**Generation of large-scale and small-scale sub-regions across Eurasia.** Since westerlies convey water vapor from NATO to the mid-latitude (N30°–N60°) Eurasia[44,45], to investigate the potential atmospheric connections of water cycle elements along the water vapor trajectories, this study first dispersed the MLE as the large-scale sub-regions and small-scale sub-regions, respectively.

For the large-scale sub-regions, based on the locations of water vapor particles in the main source regions (regions with the three largest contribution rates of moisture) over the whole simulation period (2003–2017) of FLEXPART, we initially clustered the main source regions using the $k$-means cluster method[51]. For a more detailed analysis, AS (excluding XJ), EU (including Russia) and NATO were separated as AS1-2, EU1-3, and NATO1-4, respectively. Further, for the small-scale sub-regions, the MLE was divided into 19 sub-regions (10° × 10°) along water vapor trajectories.

**Pearson correlation analysis between water cycle components.** To explore the atmospheric connections of water cycle elements between water vapor source and target regions, here, we first analyzed the consistencies between PMEs in AS1-2, EU1-3 and NATO1-4, and PMEs in the northern and southern XJs using the Pearson correlation analysis method. In addition, consistencies between TWSs in the northern and southern XJs and PMEs in the NATO1-4, TWSs in the AS1-2 and EU1-3 were evaluated to explore the impact of variations in PMEs in NATO1-4 and variations in TWSs in the AS1-2 and EU1-3 on the variations in TWSs in XJ along the water vapor particles' trajectories. The Pearson correlation analysis in this study was performed using the R package "stats".

To study the impact of variations in PMEs in NATO1-4 on the variations in TWSs in MLE, from the perspective of the small-scale analysis, we performed the cross-correlation analysis[54] between PMSs in NATO1-4 and TWSs in 19 small-scale sub-regions, which determined that variations in TWSs in 10/19 sub-regions were sensitive to the variations in PMEs in NATO3. Thus, we further defined 10 sub-regions where TWSs were highly correlated with PME in NATO3 (the cross-correlation > 0.50) as high-correlated sub-regions (HSR), and defined other sub-regions as low-correlated sub-regions (LSR). The cross-correlation analysis was performed using the R package "stats".

**Modified Mann-Kendall trend analysis of TWSs in lands and PMEs in ocean.** In this study, the modified Mann-Kendall trend analysis method was applied to study the spatiotemporal trends of water cycle elements (PMEs, SSTs and TWSs) in NATO and MLE. To compute the MMK trends, the R package "modifiedmk" was used.

**Identification of the seasonal landfalling routes of PME deficits from NATO to Eurasia.** The landfalling routes of droughts were identified by tracking the drought clusters[55,56]. To identify the landfalling routes of PME deficits originating from NATO, we first analyzed the spatiotemporal patterns of the averaged PME anomalies over NATO and averaged the TWS anomalies across Eurasia from January to December during 2003–2017, and the spatial patterns of the sums of above anomalies.

**Projection model of TWS in HSRs during 2020–2050 based on the random forest model.** The above analysis demonstrated that the variations in TWS in HSR1-10 across the MLE was sensitive and highly related to the variation in PME in NATO1 and NATO3. Thus, we trained and modified the random forest models between TWS in HSR1-10 and PME in both NATO1 and NATO3 during 2003–2017 based on ERA5 and GRACE datasets using the R package "randomForest"[57] (Eq. 12),

$$TWS_i = RandomForest(PME_{NATO_1}, PME_{NATO_3}) \qquad (12)$$

where $TWS_i$ refers to the TWS in HSR $i$; and the $PME_{NATO_1}$ and $PME_{NATO_3}$ refer to the precipitation-minus-evapotranspiration in NATO1 and NATO3, respectively. The above indices were all dispersed and regrouped randomly by 30 (validation)/70(train) as the training and modification samples to the model achieved the best performance. The performance of model was evaluated quantitatively by using root-mean square error (RMSE). Then, the projected TWS of the best model based on the PME over NATO1 and NATO3 was further validated using GRACE-based TWS during 2003–2017 from the perspective of the Pearson's correlations. Moreover, the Pearson's correlation coefficients between observed TWS and projected TWS during 2003–2017 (study period), which ranged between 0.83 and 0.91 ($p < 0.05$) after training and validation, corroborating the applicability of the random forest models in predicting TWS changes over HSR1-10 (Supplementary Fig. 16).

To evaluate the importance of PMEs in NATO1 and NATO3 for models in HSR1-10, the mean decrease Gini index (IncNodePurity) and the mean decrease accuracy (%IncMSE) were applied[57]. Based on the importance matrix of models, both the IncNodePurity and the %IncMSE demonstrated that the impact of the PME deficit over NATO3 on TWS variations over HSR1-10 was larger than that of the PME deficit over NATO1 during 2003–2017 (Supplementary Fig. 15).

Different from the ERA5-based PME (Fig. 2; and Supplementary Figs. 9 and 12), the historical multiple-models averaged CMIP6-based PME over NATO1 and NATO3 were poorly related to TWSs in HSR1-10 and the SSTs over NATO1 and

NATO3 during 2003 to 2017 (Supplementary Figs. 20–21). Thus, this study projected the future (2018–2050) spatiotemporal variations in TWSs across the MLE using the best random forest models trained based on ERA5 historical data with input as PMEs excluding SSTs (NATO1 and NATO3) derived from the CMIP6-SSP245 and CMIP6-SSP585 datasets.

**Attribution analysis for TWS variations across MLE considering interactions between total freshwater withdrawals (TFW) and PME over NATO.** We developed a linear regression model (Eqs. 13–15) to quantify the relationships between TWS, TFW and PME over NATO, evaluating fractional contributions of human activities and climate changes to TWS variations over Eurasia. The linear regression model was developed as follows.

$$\text{TWS} = a \times \text{TFW} + b \times \text{PME}_{\text{NATO}_3} \text{(declining TWS)} \quad (13)$$

$$\text{TWS} = a \times \text{TFW} + b \times \text{PME}_{\text{NATO}_3} + c \times \text{PME}_{\text{NATO}_4} \text{(increasing TWS)} \quad (14)$$

$$\text{diff} = \text{abs}(a) - (\text{abs}(b) + \text{abs}(c)) \quad (15)$$

where $a$, $b$, and $c$ are the coefficients for TFW and PME over NATO3 and PME over NATO4, respectively. For most regions across MLE where TWS had declining trends, the combined impacts from the variations in TFW and declines in PME over NATO3 were evaluated using Eq. 13. For regions where TWS showed increasing trends and was spatially close to NATO4, the combined impacts from variations in TFW, decline in PME over NATO3, and increase in PME over NATO4 were evaluated using Eq. 14. The difference between the coefficients was estimated using Eq. 15. A negative diff indicated that TWS variation was dominated by the influence from TFW (TFW-dominated TWS), while a positive *diff* indicated that TWS variation was dominated by the variations in PME over NATO (PME-NATO-dominated TWS). Since the impact of TFW on the TWS variation was negative, the computation with $a > 0$ was not included in the attribution analysis. However, we identified an interesting phenomenon behind the positive $a$, which is discussed with details in Supplementary Text 3.

**Maximum covariance analysis between PME over NATO and TWS across Eurasia.** We performed the maximum covariance analysis (MCA) to identify the leading modes of shifting linkages between PME and TWS (Eqs. 16–20). The computational demand for the MCA analysis to specifically solve the spatial matrix was substantial, which we addressed by employing graphics processing unit (GPU).

$$\mathbf{X} = \begin{bmatrix} \mathbf{X}_1(1) & \cdots & \mathbf{X}_1(N) \\ \vdots & \ddots & \vdots \\ \mathbf{X}_m(1) & \cdots & \mathbf{X}_m(N) \end{bmatrix} \quad (16)$$

$$\mathbf{Y} = \begin{bmatrix} \mathbf{Y}_1(1) & \cdots & \mathbf{Y}_1(N) \\ \vdots & \ddots & \vdots \\ \mathbf{Y}_q(1) & \cdots & \mathbf{Y}_q(N) \end{bmatrix} \quad (17)$$

$$\mathbf{C}_{x,y} = \frac{1}{N} \mathbf{X} \mathbf{Y}^T = \mathbf{U} \begin{bmatrix} \Sigma & 0 \\ 0 & 0 \end{bmatrix} \mathbf{V}^T \quad (18)$$

$$\text{PC}_{x,m} = \mathbf{U}_m^T \mathbf{X} \quad (19)$$

$$\text{PC}_{y,q} = \mathbf{V}_q^T \mathbf{Y} \quad (20)$$

where $\mathbf{X}$, $\mathbf{Y}$ refer to the spatial matrix of TWS across Eurasia and PME over NATO with N column as time span, respectively. $m$ and $q$ refer to the number of cells for the spatial matrix of TWS and PME, respectively. $\mathbf{C}_{x,y}$ is the covariance matrix. $\mathbf{U}$ and $\mathbf{V}$ are the spatial modes for TWS and PME, respectively. $\text{PC}_{x,m}$ and $\text{PC}_{y,q}$ are the temporal sequence for cell $m$ of TWS and cell $q$ of PME.

## Data availability
All data used in this study were obtained from the public database. ERA-interim meteorological data ($0.75° \times 0.75°$) during the 2003–2017 were acquired and preprocessed by the flex_extract software for the simulation of moisture transport. The monthly terrestrial water storage anomaly data (2003–2017) were obtained from the GRACE RL05 Mascon dataset (http://www2.csr.utexas.edu/grace). The monthly precipitation data (2003–2017) with the spatial resolution as $0.25° \times 0.25°$ used in this study were from GPCC (Global Precipitation Climatology Centre) monthly precipitation data[58] (https://opendata.dwd.de/climate_environment/GPCC/) and ERA5 (ECMWF Reanalysis 5th generation) monthly precipitation data[59]. Monthly potential evapotranspiration (2003–2017) and sea surface temperature (2003–2017) at the spatial resolution of $0.25° \times 0.25°$ were acquired from ERA5[59]. Monthly ocean current (1993–2019) in the North Atlantic Ocean (NATO) at the spatial resolution of $0.33° \times 0.33°$ was acquired from the OSCAR (Ocean Surface Current Analysis Real-time) dataset[60]. The multi-model monthly CMIP6 precipitation, evapotranspiration and sea surface temperature (2003-2050) was acquired from CEDA (Center for the

Environmental Analysis). The CMIP6 models used in this study included ACCESS-ESM1.5, BCC-CSM2-MR, CanESM5, GFDL-ESM4, IPSL-CM6A-LR, MICRO6, MRI-ESM2.0, and NorESM2-LM. The annual total freshwater withdrawals (TFW) data at the country scale across the MLE during 2003–2016 were acquired from the Organization for Economic Co-operation and Development (OECD, https://data.oecd.org/water/water-withdrawals.htm) and the World Bank (https://data.worldbank.org/). And the annual TFW data for the Xinjiang of China and the North China Plain were sourced from the China Data Insights (https://cdi.cnki.net). The data involved in the study have been deposited in the public repository Zendo (https://doi.org/10.5281/zenodo.5842890)[61].

## Code availability
The Graphics Processing Unit (GPU) codes in R language used for the maximum covariance analysis (MCA) are available on Zenodo (https://doi.org/10.5281/zenodo.5823882)[62]. All data and codes for all figures are available on Zenodo (https://doi.org/10.5281/zenodo.5842890)[61].

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

## Acknowledgements

Z.S., Q.Z., and W.W. acknowledge the support from the China National Key R&D Program (Grant No. 2019YFA0606900), the Engineering Research Center for Water Resources & Ecological Water in Cold and Arid Region of Xinjiang Uygur Autonomous Region, China (Grant No. 2020.A-003), the National Science Foundation of China (Grant No. 41771536), and water consumption and water balance simulation and ecological water transportation research in Hetian River Basin (TGJHTJJG-2018KYXM0001). Y.P. acknowledges the support from the National Science Foundation (Grant No. 1752729).

## Author contributions

Z.S. and Q.Z. designed the research and wrote the manuscript. Z.S. performed the analysis. V.S., Y.P., J.L., C.Y.X., and W.W. discussed and modified the manuscript. All authors contributed to the interpretation of results.

## Competing interests

The authors declare no competing interests.
