## [Peer Review File · Nature Communications]

Drying in the low-latitude Atlantic Ocean contributed to terrestrial water storage depletion across EurasiaReviewers' Comments:

Reviewer #1:

Remarks to the Author:

The manuscript shows that drying in the low-latitude Atlantic Ocean attributed to terrestrial water storage depletion across Eurasia. While the manuscript presents an important work, I have two major comments that authors can address carefully:

1) It is not clear what fractional of the total observed change in TWS over Eurasia is driven by human interventions and climate. Authors need to first show the relative contribution of the three before attributing the major changes to climate.

2) The skill of CMIP6 models to simulate P-E over the lower latitude of north Atlantic ocean needs to be examined before using the models for the future projections.

3) Physical mechanism of the contribution of the changes in P-E over the ocean to TWS changes needs to be carefully presented. For instance, it remains unclear what drives the changes in P-E and how those are linked with the TWS changes in the regions. I would suggest using Maximum Covariance Analysis to identify the leading modes of variability of the linkage between P-E and TWS.

Reviewer #2:

Remarks to the Author:

This study investigated teleconnections between drying in low-latitude North Atlantic Ocean (LNATO) and terrestrial water storage (TWS) depletions across mid-latitude Eurasia (MLE). By using the Lagrangian transport and dispersion model, detecting correlations between decreased TWS in MLE and the decreased precipitation-minus-evapotranspiration (PME) in LNATO, this study provided an alternative view in compare with atmospheric circulation modeling. It declared the the routines and contribution of northeastward propagation of PME deficits from LNATO induced TWS depletion in MLE both during 2003-2017 and in the future period during 2031-2050.

TWS changes in Eurasia could be driven by both climatic variations and human activities, among which the anthropogenic activities have been suggested to be among the key factors causing TWS loss, for example the decreased TWS in the Northwest China (Xinjiang), the North China Plain, the Middle East and the Southwest Russia during 2002-2017 attributed by agricultural irrigation, mining dewatering and/or domestic water withdrawals. It is also known that the role of teleconnection is more limited than for the precipitation, as also found in this study that the long-term averaged CR for NATO were 2.8%. Clear seasonal differences in the moisture sources exits between wet and dry seasons. The main issues is that the authors should more clearly demonstrate by the analysis the extent in space and time the drying in the low-latitude Atlantic Ocean attributed to terrestrial water storage depletion across Eurasia. Therefore, the reviewer suggests the major revision before the consideration of the publication. The specific comments are as followed.

1. Line 54-56: TWS is almost the sole freshwater resource beside precipitation, no matter it is arid or humid region.

2. Line 72-79 Please give more specific and detailed informations on the spatial and temporal extent of the TWS change driven mainly by climatic forcing.

3. Line 91-92, How much of the attribution, in which season?

4. Line 158-160. It is vague. The water vapor source for AS and EU is variable in time.

5. In the section "Linkage between water vapor over NATO and TWS across MLE" and throughout the manuscript, please marked the relation between PME an TWS with the statistical significance of the correlation analysis.

6. Line 182-183. How you consider time lag in defining eastward and landward water vapor transport

route.

7. Figure 2 What the 'a,b,c....j' stand for....

8. Line 310-312, Please update the descriptions.

9. Line 314, TWS decrease doesn't surely mean water scarcity.

10. Low-latitude NATO is consistent with the decrease of TWS along the HSR1-10, while high- latitude NATO shows less or uncorrelated. Further explanations are needed for the reason.

11. Line 317 The findings indicate that atmospheric drivers have played broader TWS declines across MLE, however evidence is lack to show it is the key role.

12. Too many abbreviations add the difficulty in understanding the manuscript

Reply to Reviewer #1 (Remarks to the Author):

The manuscript shows that drying in the low-latitude Atlantic Ocean attributed to terrestrial water storage depletion across Eurasia. While the manuscript presents an important work, I have two major comments that authors can address carefully:

1) It is not clear what fractional of the total observed change in TWS over Eurasia is driven by human interventions and climate. Authors need to first show the relative contribution of the three before attributing the major changes to climate.

Reply: We thank the reviewer for the professional suggestions and constructive comments, which helped tremendously in improving the quality of the manuscript. We have thoroughly revised the manuscript, carefully addressing all comments from the reviewer.

We note that most of the new figures we added during the revision have been presented in the Supplementary Information due to space limitation in the main text.

The address the comment regarding fractional terrestrial water storage (TWS) change caused by human intervention and climate variability, we obtained annual total freshwater withdrawal (TFW) data at a national/region scale across the mid-latitude Eurasia during 2003-2016. These data were sourced from the Organization for Economic Co-operation and Development (OECD, <https://data.oecd.org/water/water-withdrawals.htm>) and the World Bank (<https://data.worldbank.org/>). The annual TFW data for the Xinjiang of China and the North China Plain were obtained from the China Data Insights (<https://cdi.cnki.net>). Here, the TFW was defined as the total freshwater withdrawals by domestic, agricultural and industrial activities.

Then, we developed a linear regression model (Eqs. 1-3) evaluating relations between TWS, TFW and PME (precipitation-minus-evapotranspiration) over the NATO, quantifying fractional contributions of human activities and climate changes to variations in TWS over Eurasia. The absolute coefficients for each factor were used to evaluate attributions of each factor to variations in TWS. The linear regression model and related coefficients were introduced as follows:

$$TWS = a \times TFW + b \times PME_{NATO3} \text{ (declining TWS)} \quad (\text{Eq. 1})$$

$$TWS = a \times TFW + b \times PME_{NATO3} + c \quad (\text{Eq. 2})$$

$$\begin{aligned} & \times PME_{NATO4} \text{ (increasing TWS)} \\ \text{diff} = \text{abs}(a) - (\text{abs}(b) + \text{abs}(c)) \quad (\text{Eq. 3}) \end{aligned}$$

where a, b and c are the coefficients for TFW and PME over NATO3 and PME over NATO4, respectively. For most regions across MLE where TWS had declining trends, the combined impacts from the variations in TFW and declines in PME over NATO3 were evaluated using Eq.1. For regions where TWS showed increasing trends and was spatially close to NATO4, the combined impacts from variations in TFW, decline in PME over NATO3, and increase in PME over NATO4 were evaluated using Eq. 2. Difference between the coefficients was estimated using Eq. 3. A negative diff

indicated that TWS variation was dominated by the influence from TFW (TFW-dominated TWS), while a positive diff indicated that TWS variation was dominated by the variations in PME over NATO (PME-NATO-dominated TWS). Since the impact of TFW on TWS variation is negative, the computation with $a > 0$ was not included in the attribution analysis. However, we identified an interesting phenomenon behind the positive a , which is discussed with details in Supplementary Text 3. Out of your convenience, the detailed discussion has been listed as follows.

Reply Fig. 1a-d indicates that TWS in 71% of the regions with negative a coefficient was influenced mainly by variations in PME over the NATO; while TWS in 29% of these regions was influenced mainly by TFW. Only 5 out of the regions with decreasing TWSs were featured by TFW-dominated TWS such as Bosnia and Herzegovina (BosHerz for abbreviation in plot), Lebanon, Pakistan, Tunisia and Xinjiang (China) (Reply Fig. 1e). Despite of the differences in variations of TFW, the decrease in TWS in these regions were in good line with the decrease PME over the NATO3 (Reply Fig. 1g-k). It should be emphasized that the variations in TWS in Bosnia and Herzegovina, Tunisia and Xinjiang (China) are highly sensitive to abrupt changes in PME over NATO3 in 2015 (Reply Fig. 1g-k). Meanwhile, when compared to regions with increasing TWS, the regions with decreasing TWS mainly distributed along the eastward propagation route of the landfalling PME-deficit originated from the low-latitude NATO. The temporal variation and spatial pattern both demonstrated that the PME deficit originated from the low-latitude NATO induced widespread decrease in TWS across the mid-latitude Eurasia. For the sake of drought mitigation in abovementioned 5 regions, TFW can exacerbate the deficit of TWS in the mid-latitude Eurasia.

France, Netherlands, Spain and Sweden were the 4 regions with increasing TWSs which were featured as TFW-dominated TWS (Reply Fig. 1f). However, the declines in TFW in France, Netherlands and Spain did not significantly restore the TWS during 2003-2009 and the increase in TFW in Sweden did not significantly lead to the decline in the TWS during 2003-2009 (Reply Fig. 1l-o). Despite of the spatial heterogeneity of TFW over regions with increasing TWSs, the TWS over above-mentioned 4 regions was in good line with variation in PME over the NATO4 during 2003-2015. Besides, different from the regions with the declining TWS, the regions featured by the increasing TWS distributed mostly in the western, southern and northern Europe, where is spatially close to the NATO4 (Reply Fig. 1f). In summary, it demonstrated that the increase in PME over the NATO4 directly triggered increasing trends in TWS while decline in TFWs did not significantly impact the TWS over the western, southern and northern Europe during 2003-2016.

Reply Figure 1. (Supplementary Figure 18). Attribution analysis for TWS variation across the Eurasia at the regional/national scale. (a-c) refer to the coefficients for the TFW, PME over the NATO3 and PME over the NATO4, respectively. (d) refers to the difference between the absolute value of the coefficient a and sum of the absolute values of the coefficient b and c . The negative difference value indicates that TWS variation is influenced mainly by the TFW (TFW-dominated TWS); while the positive difference value indicates that TWS variation is influenced mainly by PME variation over the NATO (PME-NATO-dominated TWS). (e-f) refer to the spatial patterns of the regions with TFW-dominated TWS or PME-NATO-dominated TWS with decreasing (e) or increasing (f) TWS. (g-o) refer to the temporal variations in TWS, TFW, PME-NATO3 and PME-NATO4 at regions with TFW-dominated TWS during 2003-2016.

With exception of human impacts on water storage, it is interesting to find influence of variation in water storage on freshwater withdrawal behavior at regional scale, which is reflected by the positive coefficient a , implying synchronous changes between TFW and TWS (Reply Fig. 2). It violated the assumption that TFW has

negative impacts on TWS. Meanwhile, we evaluated relations between TWS and TFW and found that the correlation coefficient is > 0.3 over 65% of these regions (Reply Fig. 2), such as Belgium, Denmark, Greece, Poland, Israel, Slovenia, Georgia, Iraq, Kyrgyzstan, Kazakhstan, Moldova, Montenegro, Syria, Tajikistan and Ukraine. Moreover, variations in TWS were in agreement with variations in TFW and the increasing rates of GDP (gross domestic production) (Reply Fig. 2c-q). Of course, it makes no sense if we attempt to elucidate these phenomena from the perspective of the negative impacts of the local economic development and TFW on TWS. Instead, these phenomena demonstrated that variation in TWS could impact the freshwater withdrawal behavior at regional scale, and further influence the local economic development due to constraint of the freshwater withdrawals.

In summary, we compared impacts of TFW and PME over the NATO on TWS at regional scale, and evaluated interactions between TFW and TWS. The TWS across the mid-latitude Eurasia was dominated by PME over the NATO in 71% of the regions and was mainly influenced by TFW in 29% of the regions. Besides, we also demonstrated that variation in TWS could impact the freshwater withdrawal behavior at regional scale, and could further influence the local economic development.

Reply Figure 2. (Supplementary Figure 19). The impact of decline in TWS across the mid-latitude Europe on the freshwater withdrawal behavior and local economic development. (a) Spatial pattern of the modified Mann-Kendall trends of the TWS and relations between TWS and TFW at national/regional scale; (b) Statistic analysis of the relations between TWS and TFW at national/regional scale. Red filled circles marked by white cross show significant relations at 95%. (c-q) Temporal variations in TWS, TFW and growing magnitude of Gross Domestic Product (GDP_GR). The GDP_GR data for Syria is not full.

2) The skill of CMIP6 models to simulate P-E over the lower latitude of north Atlantic ocean needs to be examined before using the models for the future projections.

Reply: Thank you for your insightful review and your professional suggestions. Given the uncertainties in the CMIP6 models in simulating the P-E over the lower latitude of North Atlantic Ocean, we trained and tested the models for the projection of TWS using P-E based on the historical ERA5 dataset, and projected future variation in TWS based on the original P-Es by CMIP6 models. To corroborate and confirm reliability of projection results, we evaluated the skill or modelling performance of CMIP6 models in simulating P-E over the lower latitude of North Atlantic Ocean, and further projected TWS with future P-E based on the multiple weighted CMIP6 ensemble (Supplementary Text 2).

At first, we examined the skill of CMIP6 models in simulating P-E over the lower latitude of North Atlantic Ocean in comparison with the ERA5-PME during the 2003-2014. According to the Reply Figs. 3-4, the relations between ERA5-PME and CMIP6-PMEs varied from model to model. For the simulation of the PME in NATO1, the absolute value of the correlation coefficients between ERA5-PME and CMIP6-PMEs ranged from 0.05 to 0.49. With exception of PME simulated by IPSL-CM6A-LR and MRI-ESM2.0, the rest of the simulated CMIP6-PMEs over NATO1 were all significantly correlated with ERA5-PME at 95% significance level. For the simulated PME in NATO3, the absolute value of the correlation coefficients between ERA5-PME and CMIP6-PMEs ranged from 0.01 to 0.16. With exception of the PME simulated by IPSL-CM6A-LR and MRI-ESM2.0, the rest of the simulated CMIP6-PMEs over NATO3 were all significantly correlated with the ERA5-PME at 95% significance level. Despite low correlation coefficients between the CMIP6-PME and ERA-PME over the NATO3, the simulated PME by some of the CMIP6 models could accurately simulate the maximum and minimum PME during 2003-2014.

Reply Figure S3 (Supplementary Figure 22). Evaluation of the CMIP6-PME over the NATO1 in comparison with the ERA5-PME during 2003-2014.

Reply Figure S4 (Supplementary Figure 23). Evaluation of the CMIP6-PME over the NATO3 in comparison with the ERA5-PME during 2003-2014.

To improve the accuracy of the simulated CMIP6-PME over the NATO, we developed initial linear regression models quantifying relations between ERA5-PME and CMIP6-PME during 2003-2017 to determine the weights for the calculation of the multiple CMIP6 model ensemble. Since the historical periods for the CMIP6 models end in 2014, we developed the linear regression models for historical-SSP245 and historical-SSP585 CMIP6 models during 2003-2017, respectively.

$$PME_{ERA5} = \sum_{i=1}^8 a_i \times PME_i \quad (\text{Eq. 4})$$

where PME_{ERA5} refers to the ERA5-PME over the NATO and PME_i refers to the simulated PME by the CMIP6 model i . a_i refers to the linear regression coefficients for the PME_i . The CMIP6 models 1-8 include ACCESS-ESM1.5, BCC-CSM2-MR, CanESM5, GFDL-ESM4, IPSL-CM6A-LR, MIROC6, MRI-ESM2.0 and NorESM2-LM.

Comparison between the linear regression coefficients and significance levels among the simulated PMEs by CMIP6 models indicated that PMEs simulated by ACCESS-ESM1.5, BCC-CSM2-MR and CanESM5 under historical-SSP245 scenario and PMEs simulated by ACCESS-ESM1.5, CanESM5 and MIROC6 under historical-SSP585 scenario were further selected to determine the weights for the calculation of the PME based on the multiple CMIP6 ensemble over the NATO1 (Eqs. 5-6). PMEs simulated by ACCESS-ESM1.5, IPSL-CM6A-LR and NorESM2-LM over the NATO3 under historical-SSP245 scenario and PMEs simulated by

BCC-CSM2-MR, CanESM5 and MIROC6 under historical-SSP585 scenario were further selected to determine the weights for the calculation of the PME based on the multiple CMIP6 over the NATO3 (Eqs. 7-8).

$$PME_{ERA5} = \sum_{i=1,2,3} a_i \times PME_i \text{ (over NATO1 under Historical - SSP245)} \quad (\text{Eq. 5})$$

$$PME_{ERA5} = \sum_{i=1,3,6} a_i \times PME_i \text{ (over NATO1 under Historical - SSP585)} \quad (\text{Eq. 6})$$

$$PME_{ERA5} = \sum_{i=1,5,8} a_i \times PME_i \text{ (over NATO3 under Historical - SSP245)} \quad (\text{Eq. 7})$$

$$PME_{ERA5} = \sum_{i=2,3,6} a_i \times PME_i \text{ (over NATO3 under Historical - SSP585)} \quad (\text{Eq. 8})$$

Reply Figs. 5-6 illustrated that the simulated PMEs by the weighted CMIP6 models ensemble over the NATO1 and NATO3 were all significantly correlated with the ERA5-PMEs under the historical-SSP245 and historical-SSP585 scenarios during 2003-2016. The correlation coefficients between ERA5-PME and simulated CMIP6-PME over the NATO1 has been improved from 0.05-0.49 to 0.4-0.6. And the correlation coefficients between ERA5-PME and simulated CMIP6-PME over the NATO3 has been improved from 0.01-0.16 to 0.44-0.56.

For the sake of the credibility of the projected TWS under SSP245 and SSP585 scenarios, we calculated the bias-corrected PMEs over the NATO1 and NATO3 based on the weighted CMIP6 model ensemble, respectively (Reply Fig. 7), and further projected the bias-corrected TWS using the trained model based on the historical TWS and ERA5-PME (Reply Fig. 8). During 2018-2050, the projected TWS based on the original simulation of the CMIP6 models and the bias-corrected TWS based on the weighted CMIP6 models ensemble both showed similar variations of TWS across the Eurasia under the SSP585 scenarios (Reply Fig. 8b). However, under SSP245 scenario, the originally projected TWS firstly increased by 111% while bias-corrected TWS decreased by 291% during 2018-2031, and both declined during 2031-2050 (Reply Fig. 8a).

Reply Figure 5 (Supplementary Figure 24). Evaluation of the simulated PME by the weighted CMIP6 models ensemble over the NATO1 under historical-SSP245 and historical-SSP585 scenarios. (a-b) refer to the temporal variations in ERA5-PME and simulated PMEs based on the 80% of the randomly sampled training dataset under historical-SSP245 and historical-SSP585 scenarios. (c-d) refer to the temporal variations in ERA5-PME and simulated PMEs based on the 20% of the randomly sampled testing dataset under historical-SSP245 and historical-SSP585 scenarios. (e-f) refer to the temporal variations in ERA5-PME and simulated PMEs by the weighted CMIP6 models ensemble during 2003-2017.

Reply Figure 6 (Supplementary Figure 25). Evaluation of the simulated PME_s by the weighted CMIP6 models ensemble over the NATO3 under historical-SSP245 and historical-SSP585 scenarios. (a-b) refer to the temporal variations in ERA5-PME and simulated PME_s based on the 80% of the randomly sampled training dataset under historical-SSP245 and historical-SSP585 scenarios. (c-d) refer to the temporal variations in ERA5-PME and simulated PME_s based on the 20% of the randomly sampled testing dataset under historical-SSP245 and historical-SSP585 scenarios. (e-f) refer to the temporal variations in ERA5-PME and simulated PME_s by the weighted CMIP6 models ensemble during 2003-2017.

Reply Figure 7 (Supplementary Figure 26). Comparisons between the projected PME based on the original simulation of the CMIP6 models (Proj_PME) and weighted CMIP6 models ensemble (BC_Proj_PME) over the NATO1 and NATO3 under SSP245 and SSP585 scenario during 2018-2050.

Reply Figure 8 (Supplementary Figure 17). Comparisons between the projected TWS across the mid-latitude Eurasia based on the projected PME by the original simulation of the CMIP6 models (Proj_TWS) and weighted CMIP6 models ensemble (BC_Proj_TWS) over the NATO1 and NATO3 under SSP245 and SSP585 scenario during 2018-2050.

3) Physical mechanism of the contribution of the changes in P-E over the ocean to TWS changes needs to be carefully presented. For instance, it remains unclear what drives the changes in P-E and how those are linked with the TWS changes in the regions. I would suggest using Maximum Covariance Analysis to identify the leading modes of variability of the linkage between P-E and TWS.

Reply: Thank you cordially for your professional suggestions and pertinent comments. As for your first question “it remains unclear what drives the changes in P-E”, we have discussed it in the previous version of our manuscript. Here we also presented discussions as follows. However, the core issue we attempted to address is to delve into the connections between P-E over the NATO and the TWS across the Eurasia. Moreover, we devoted to anatomize possible driving factors and physical mechanisms behind connections between P-E over the NATO and the TWS across the Eurasia. However, your question “it remains unclear what drives the changes in P-E” is very interesting and we will go on with this question in our ongoing investigation. Thank you so much for your insightful review and your professional comments.

The relative discussion on the first question “it remains unclear what drives the changes in P-E” is as follow. The SST variations could be one of the driving factors causing PME deficits over NATO^{47,48}. Here we evaluated the relation between SST and PME during 2003-2017 using the ERA5 reanalysis historical data. We found that declining SST within N0°~N20° NATO (the modified Mann-Kendall (MMK) trends ranged between (-4, 0]) and slightly increasing SST within N20°~N40° NATO

(MMK trends ranged between (0, 2]) resulted in the decreased PME (MMK trends ranged between (-2, 0]) (Supplementary Figs. 12a, b, c, e, g). The drastic warming SST (MMK trends ranged between (2, 8]) tended to increase PME over NATO1. In general, the SST variations were positively correlated with the PME changes over low-latitude ($N0^{\circ}\sim N40^{\circ}$) NATO. This finding is consistent with previous studies^{47,48}. Besides, we found reversed relations between SST and PME over high-latitude NATO when compared to LNATO (Supplementary Fig. 12h). The drastic cooling SST (MMK trends ranged between (-6, 0]) steadily increased PME over $N40^{\circ}\sim N80^{\circ}$ NATO (Supplementary Figs. 12a, b, f, h).

47. Xie, S., et al. Global warming pattern formation: Sea surface temperature and rainfall. *J. Clim.* 23(4), 966-986(2010).

48. He, J., & Soden, B. J. A re-examination of the projected subtropical precipitation decline. *Nat. Clim. Chang.* 7(1), 53-57 (2017).

Supplementary Figure 12 Spatial patterns of the modified Mann-Kendall (MMK) trends of the sea surface temperature (SST, a) and precipitation-minus-evapotranspiration (PME, b) in northern Atlantic Ocean (NATO). Temporal covariations between normalized trend items of regional averaged SST in NATO1-4 and normalized trend items of regional sums of PMEs in NATO1-4 (c-f), the normalized trend item of regional sums of PMEs in NATO1 and NATO3(g), the

normalized trend item of regional sums of PME in NATO2 and NATO4(h), respectively.

As for your second question “how variation in P-E over the NATO was linked with the TWS changes in the regions”. It is an important issue and we did the maximum covariance analysis (MCA) to identify the leading modes of variability of the linkage between PME and TWS (Eqs. 9-13). Besides, given the massive computation workload between spatial matrix, we have improved the MCA analysis procedure using the graphics processing unit (GPU), and further we would upload the MCA-GPU calculation codes in R language to the Github (DOI: <https://doi.org/10.5281/zenodo.5823882>).

$$X = \begin{bmatrix} X_1(1) & \cdots & X_1(N) \\ \vdots & \ddots & \vdots \\ X_m(1) & \cdots & X_m(N) \end{bmatrix} \quad (\text{Eq. 9})$$

$$Y = \begin{bmatrix} Y_1(1) & \cdots & Y_1(N) \\ \vdots & \ddots & \vdots \\ Y_q(1) & \cdots & Y_q(N) \end{bmatrix} \quad (\text{Eq. 10})$$

$$C_{xy} = \frac{1}{N}XY^T = U \begin{bmatrix} \Sigma & 0 \\ 0 & 0 \end{bmatrix} V^T \quad (\text{Eq. 11})$$

$$PC_{x,m} = U_m^T X \quad (\text{Eq. 12})$$

$$PC_{y,q} = V_q^T Y \quad (\text{Eq. 13})$$

where X, Y refer to the spatial matrix of the TWS across the Eurasia and PME over the NATO with N column as time span. m and q refer to the number of cells for the spatial matrix of the TWS and PME. C_{xy} is the covariance matrix. U and V are the spatial modes for the TWS and PME, respectively. $PC_{x,m}$ and $PC_{y,q}$ are the temporal sequence for the cell m of TWS and cell q of PME.

It can be seen from Reply Fig. 9 and Supplementary Text 1 that the explained variances of the leading modes 1-4 are 28.72%,13.09%, 9.29% and 7.12%, respectively. And the temporal changes of PME and TWS of the leading modes 1-4 are highly correlated with the correlation coefficients ranging between 0.86-0.94 at 95% significance level. It verified that the changing tendencies in PME and TWS were in high match and coherency (Reply Figs. 9a, d, g, j). Particularly, in the first leading mode, the changing tendency in PME over the low-latitude NATO was in good line with that of TWS across the mid-latitude Eurasia (Reply Figs. 9a-c). Meanwhile, the changing tendency in PME over the high-latitude NATO was in good agreement with that of TWS in southern, eastern and northern Europe and most regions of the western Russia (Reply Figs. 9a-c). This spatial pattern match well the propagation routes of the landfalling impacts of PMEs deficit over northern NATO landwards during 2003-2017 (see Figure 3b of the main text) (Reply Figs. 9a-c).

Reply Figure 9 (Supplementary Figure 14). The maximum covariance analysis (MCA) between the TWS across the Eurasia and PME over the NATO. (a, d, g, j) are the temporal variations of TWS and PME in the leading modes 1-4. (b, e, h, k) are the spatial modes of PME in the leading modes 1-4. (c, f, i, l) are the spatial modes of TWS in the leading modes 1-4.

Figure 3. Longitudinal profile of the monthly-averaged precipitation-minus-evapotranspiration (PME) anomalies in the mid-latitude (N30°-N60°) Atlantic Ocean (NATO) and the monthly mean terrestrial water storage (TWSs) anomalies in mid-latitude (N30°-N60°) North Africa (NAF) and Eurasia from January to December (a). Spatial patterns of the sum of monthly mean PME anomalies (unit: mm) in NATO and the sum of negative monthly mean TWS anomalies (unit: mm) in lands during 2003-2017 (b). Propagation routes of the landfalling impacts of deficits in PME over northern NATO from ocean to land during 2003-2017 (c). The NXJ, SXJ, TM, HC, LP and NCP in subplot c refer to North Xinjiang, South Xinjiang, Tian Mountain, Hexi Corridor, Loess Plateau and North China Plain, respectively.

Reply to Reviewer #2 (Remarks to the Author):

This study investigated teleconnections between drying in low-latitude North Atlantic Ocean (LNATO) and terrestrial water storage (TWS) depletions across mid-latitude Eurasia (MLE). By using the Lagrangian transport and dispersion model, detecting correlations between decreased TWS in MLE and the decreased precipitation-minus-evapotranspiration (PME) in LNATO, this study provided an alternative view in compare with atmospheric circulation modeling. It declared the routines and contribution of northeastward propagation of PME deficits from LNATO induced TWS depletion in MLE both during 2003-2017 and in the future period during 2031-2050.

TWS changes in Eurasia could be driven by both climatic variations and human activities, among which the anthropogenic activities have been suggested to be among the key factors causing TWS loss, for example the decreased TWS in the Northwest China (Xinjiang), the North China Plain, the Middle East and the Southwest Russia during 2002-2017 attributed by agricultural irrigation, mining dewatering and/or domestic freshwater withdrawals.

Reply: We thank the reviewer for the professional suggestions and constructive comments, which helped tremendously in improving the quality of the manuscript. We have thoroughly revised the manuscript, carefully addressing all comments from the reviewer. Specifically, we performed attribution analysis for the variation in TWS considering interactions between total freshwater withdrawal (TFW) and variations in PME over NATO. More details regarding our response to each of the comments and revisions we made in the manuscript and supplementary information are presented in the following.

We note that most of the new figures we added during the revision have been presented in the Supplementary Information due to space limitation in the main text.

It is also known that the role of teleconnection is more limited than for the precipitation, as also found in this study that the long-term averaged CR for NATO were 2.8%. Clear seasonal differences in the moisture sources exits between wet and dry seasons. The main issues is that the authors should more clearly demonstrate by the analysis the extent in space and time the drying in the low-latitude Atlantic Ocean attributed to terrestrial water storage depletion across Eurasia. Therefore, the reviewer suggests the major revision before the consideration of the publication. The specific comments are as followed.

Reply: Thank you for your insightful review and for your professional comments and revision suggestions. In the former calculation of the contribution rates (CR) of water vapor for the source regions, we excluded the variation (recharge or discharge) in water vapor along the propagation routes of the water particles. Here, we attached the contribution rates of water vapor for all source regions by considering the variation in the water vapor along the propagation routes to comprehensively illustrate the linkage

between the water vapor source regions and water vapor target region. According to the Reply Fig. 10, the contribution rates of water vapor from the NATO to XJ is 8.3% with consideration of the variation in water vapor along the propagation routes.

Reply Figure 10. Contribution rates of water vapor for all source regions with consideration of the variation in the water vapor along the propagation routes. All source regions include Asia (AS), Europe (EU), North Atlantic Ocean (NATO), North Africa (NAF), the Caspian Sea (CS), the Mediterranean Sea (MS), the Arctic Ocean (AO), the Black Sea (BS), the Indian Ocean (IO), North America (NAM), the Red Sea (RS), Xinjiang (XJ), and the Pacific Ocean (PO), respectively.

Meanwhile, we also did the maximum covariance analysis to identify the variability of the linkage between PME over the NATO and TWS across the Eurasia. Reply Fig. 9 showed that the explained variances of the leading modes 1-4 are 28.72%, 13.09%, 9.29% and 7.12%, respectively (Supplementary Text 1). The temporal sequences of PME and TWS in the leading modes 1-4 are highly related with the correlation coefficients ranging between 0.86-0.94 ($p < 0.05$). It verified that the changing tendencies in PME and TWS were in remarkable coherency (Reply Fig. 9a, d, g, j). Especially, in the first leading mode, the changing tendencies in PME over the low-latitude NATO were in good line with the changing tendencies in TWS across the mid-latitude Eurasia (Reply Fig. 9a-c). The changing tendencies in PME over the high-latitude NATO were in good line with the changing tendencies in TWS in the southern, eastern and northern Europe and most areas of the western Russia (Reply Fig. 9a-c). This spatial pattern agrees with the propagation routes of the landfalling impacts of deficits in PMEs over northern NATO from ocean to land during 2003-2017 in Figure 3b of the main text (Reply Fig. 9a-c).

Reply Figure 9 (Supplementary Figure 14). The maximum covariance analysis (MCA) between TWS across the Eurasia and PME over the NATO. (a, d, g, j) are the temporal sequences for TWS and PME in the leading modes 1-4. (b, e, h, k) are the spatial modes of PME in the leading modes 1-4. (c, f, i, l) are the spatial modes of TWS in the leading modes 1-4.

Figure 3. Longitudinal profile of monthly-averaged mean precipitation-minus-evapotranspiration (PME) anomalies in the mid-latitude (N30°-N60°) Atlantic Ocean (NATO) and the monthly mean terrestrial water storage (TWSs) anomalies in mid-latitude (N30°-N60°) North Africa (NAF) and Eurasia from January to December (a). Spatial patterns of the sum of monthly mean PME anomalies (unit: mm) in NATO and the sum of negative monthly mean TWS anomalies (unit: mm) in lands during 2003-2017 (b). Propagation routes of the landfalling impacts of deficits in PME over northern NATO from ocean to land during 2003-2017 (c). The NXJ, SXJ, TM, HC, LP and NCP in subplot c refer to North Xinjiang, South Xinjiang, Tian Mountain, Hexi Corridor, Loess Plateau and North Chain Plain, respectively.

1. Line 54-56: TWS is almost the sole freshwater resource beside precipitation, no matter it is arid or humid region.

Reply: Thank you for your kind review and professional suggestions. By firmly following your revision suggestion, we revised the statement in the main text. For your reference, here we listed the revised statement as follows.

Moreover, TWS is almost the sole freshwater resource in arid regions, supporting domestic, industrial, and agricultural water demands.

2. Line 72-79 Please give more specific and detailed information on the spatial and temporal extent of the TWS change driven mainly by climatic forcing.

Reply: Thank you for your kind review and professional suggestions. Since here is the brief introduction of the research background, we have attached the results of the specific and detailed information on the spatial and temporal extent of the TWS change driven mainly by climatic forcing in Section 4 “Projected TWS variations across MLE during 2018-2050” and in the Supplementary Text 3. For your convenience, we have listed the detailed revisions as follows.

In the main text:

An attribution analysis indicated that TWS changes over 71% of the regions—where the total freshwater withdrawals (TFW) had negative impacts on TWS variations—were influenced mainly by PME variations over NATO, whereas TWS changes over 29% of these regions were impacted primarily by TFW (Supplementary Fig. 18). Besides, an evaluation of interactions between TFW and TWS suggested that variations in TWS across the MLE can affect water withdrawals at the regional scale, influencing regional economic development (Supplementary Fig. 19 and Supplementary Text 3).

In the Supplementary Text 3:

Here, the TFW was defined as the total freshwater withdrawals by domestic, agricultural and industrial activities. Reply Figs. 1a-d indicated that TWS in 71% of the regions with negative a coefficient was influenced mainly by variations in PME over the NATO; while TWS in 29% of these regions was influenced mainly by TFW. Only 5 out of the regions with decreasing TWSs were featured by TFW-dominated TWS such as Bosnia and Herzegovina (BosHerz for abbreviation in plot), Lebanon, Pakistan, Tunisia and Xinjiang (China) (Reply Figs. 1e). Despite of the differences in variations of TFW, the decrease in TWS in these regions were in good line with the decrease PME over the NATO3 (Reply Figs. 1g-k). It should be emphasized that the variations in TWS in Bosnia and Herzegovina, Tunisia and Xinjiang (China) are highly sensitive to abrupt changes in PME over the NATO3 in 2015 (Reply Figs. 1g-k). Meanwhile, when compared to regions with increasing TWS, the regions with decreasing TWS mainly distributed along the eastward propagation route of the landfalling PME-deficit originated from the low-latitude NATO. The temporal variation and spatial pattern both demonstrated that the PME deficit originated from the low-latitude NATO induced widespread decrease in TWS across the mid-latitude

Eurasia. For the sake of drought mitigation in abovementioned 5 regions, TFW can exacerbate the deficit of TWS in the mid-latitude Eurasia.

France, Netherlands, Spain and Sweden were mere 4 out of the regions with increasing TWSs which were featured by TFW-dominated TWS (Reply Figs. 1f). However, the declines in TFW in France, Netherlands and Spain did not significantly restore the TWS during 2003-2009 and the increase in TFW in Sweden did not significantly lead to the decline in the TWS during 2003-2009 (Reply Figs. 11-o). Despite of the spatial heterogeneity of TFW over regions with increasing TWSs, the TWS over above-mentioned 4 regions was in good line with variation in PME over the NATO4 during 2003-2015. Besides, different from the regions with the declining TWS, the regions featured by the increasing TWS distributed mostly in the western, southern and northern Europe, where is spatially close to the NATO4 (Reply Figs. 1f). In summary, it demonstrated that the increase in PME over the NATO4 directly triggered increasing trends in TWS while decline in TFWs did not significantly impact the TWS over the western, southern and northern Europe during 2003-2016.

Reply Figure 1. (Supplementary Figure 18). Attribution analysis for TWS variation across the Eurasia at the regional/national scale. (a-c) refer to the coefficients for the TFW, PME over the NATO3 and PME over the NATO4, respectively. (d) refers to the difference between the absolute value of the coefficient a and sum of the absolute value of the coefficient b and c . The negative difference value indicates that TWS variation is influenced mainly by the TFW (TFW-dominated TWS); while the positive difference value indicates that TWS variation is influenced mainly by PME variation over the NATO (PME-NATO-dominated TWS). (e-f) refer to the spatial patterns of the regions with TFW-dominated TWS or PME-NATO-dominated TWS with decreasing (e) or increasing (f) TWS. (g-o) refer to the temporal variations in TWS, TFW, PME-NATO3 and PME-NATO4 at regions with TFW-dominated TWS during 2003-2016.

With exception of human impacts on water storage, it is interesting to find influence of variation in water storage on freshwater withdrawal behavior at regional scale, which is reflected by the positive coefficient a , implying synchronous changes between TFW and TWS (Reply Fig. 2). It violated the assumption that TFW has negative impacts on TWS. Meanwhile, we evaluated relations between TWS and TFW and found that the correlation coefficient is > 0.3 over 65% of these regions (Reply Fig. 2), such as Belgium, Denmark, Greece, Poland, Israel, Slovenia, Georgia, Iraq, Kyrgyzstan, Kazakhstan, Moldova, Montenegro, Syria, Tajikistan and Ukraine. Moreover, variations in TWS were in agreement with variations in TFW and the increasing rates of GDP (gross domestic production) (Reply Figs. 2c-q). Of course, it makes no sense if we attempt to elucidate these phenomena from the perspective of the negative impacts of the local economic development and TFW on TWS. Instead, these phenomena demonstrated that variation in TWS could impact the freshwater withdrawal behavior at regional scale, and further influence the local economic development due to constraint of the freshwater withdrawals.

In summary, we compared impacts of TFW and PME over the NATO on TWS at regional scale, and evaluated interactions between TFW and TWS. The TWS across the mid-latitude Eurasia was dominated by PME over the NATO in 71% of the regions and was mainly influenced by TFW in 29% of the regions. Besides, we also demonstrated that variation in TWS could impact the freshwater withdrawal behavior at regional scale, and could further influence the local economic development.

Reply Figure 2. (Supplementary Figure 19). The impact induced by the decline in TWS across the mid-latitude Europe on the freshwater withdrawal behavior of the freshwater resource and local economic development. (a) Spatial pattern of the modified Mann-Kendall trends of the TWS and relations between TWS and TFW at the national/regional scale. (b) Statistic analysis of the relations between TWS and TFW at the national/regional scale. The filled circles marked by white cross show the significant relations ($p < 0.05$). (c-q) Temporal variations in the TWS, TFW and the growing rate of the Gross Domestic Product (GDP_GR).

3. Line 91-92, How much of the attribution, in which season?

Reply: Thank you for your kind review and professional suggestions. The question you raised here is the core issue of this study. For your convenience, we have listed

the revisions here by firmly following your suggestion. We did the monthly variations of moisture contribution rates from all vapor source regions to Xinjiang during 2003-2017 (Supplementary Figure 2 as follows).

The core scientific issue the study addresses is: what are various mechanisms and pathways whereby climatic changes over the ocean impact or drive TWS variations across Eurasia, and how much of the attribution in which season.

Supplementary Figure 2. Monthly variations of moisture contribution rates from all vapor source regions to Xinjiang during 2003-2017. (a-m) refer to monthly-averaged contribution rates (MCRs) from Asia (AS), Europe (EU), North Atlantic Ocean (NATO), North Africa (NAF), the Caspian Sea (CS), the Mediterranean Sea (MS), the Arctic Ocean (AO), the Black Sea (BS), the Indian Ocean (IO), North America (NAM), the Red Sea (RS), Xinjiang (XJ), and the Pacific Ocean (PO), respectively.

4. Line 158-160. It is vague. The water vapor source for AS and EU is variable in time.

Reply: Thank you for your kind review and professional suggestions. We have revised the description in the main text as follows.

However, even though NATO is the one of the important water vapor sources for AS and EU^{43,44}, due to water vapor loss during the water vapor transport, the CR of NATO was smaller than the CRs of AS and EU.

Besides, we have verified the significant impacts of the variation in the PME over the NATO on the variation in TWS across the mid-latitude Eurasia using the attribution

analysis and maximum covariance analysis and the results were presented in this document, which could demonstrate that the NATO is one of the important water vapor sources for the AS and EU. Hopefully, this reply could satisfy your requirement.

5. In the section “Linkage between water vapor over NATO and TWS across MLE” and throughout the manuscript, please marked the relation between PME and TWS with the statistical significance of the correlation analysis.

Reply: Thank you for your kind review and your professional suggestions. It is done. For your convenience, we have listed the specific revisions as follows.

Supplementary Figure 2 Temporal covariations among the normalized trend items of the regional sums of terrestrial water storages (TWSs) in sub-regions 1-10 (HSRs 1-10) across mid-latitude (N30°-N60°) Eurasia and the normalized trend items of the regional sums precipitation-minus-evapotranspiration (PMEs) in NATO1 and NATO3 (a-e and h-l, respectively). Spatial patterns of the modified Mann-Kendall (MMK) trends of PMEs in the NATO and TWSs in North Africa, Europe and Asia (g). Cross Pearson's correlation between PMEs in northern NATO and TWSs in HSRs 1-10. PME_NATO_MMK refers to the MMK trend in PMEs over NATO. TWS_MMK refers to the MMK trend in TWS across Eurasia and North Africa. The ocean currents in the plot were the long-term averaged ocean currents in NATO based on the OSCAR (Ocean Surface Current Analysis Real-time) dataset during 1993-2019. The moisture trajectories in subplot (g) were clustered using the k-means cluster method based on the locations of particles in the trajectories.

Supplementary Figure 3. Pearson correlation between normalized trend items of the zonal sums of the precipitation minus evapotranspiration (PME for abbreviation) in north (south) Xinjiang and normalized trend items of the sums of regional PME in sub-regions of the northern Atlantic Ocean (NATO1-4, a-b), Europe (EU1-3, c-d), and Asia (AS1-2, e-f), respectively.

Supplementary Figure 6. Pearson correlation between normalized trend items of the zonal sum of terrestrial water storage (TWS) in north (south) Xinjiang and normalized trend items of the zonal sums of precipitation-minus-evapotranspiration (PMEs) in sub-regions of the northern Atlantic Ocean (NATO1-4, a-b), normalized trend items of the zonal sums of TWSs in sub-regions of Europe (EU1-3, c-d) and Asia (AS1-2, e-f), respectively.

Supplementary Figure 9 Cross correlations between normalized trend items of the zonal sums of precipitation-minus-evapotranspiration (PMEs) in NATO1-4 and normalized trend items of the zonal sums of terrestrial water storages (TWSs) in smaller sub-regions (SRs) across the mid-latitude (N30° - N60°) Eurasia along the trajectories of water vapor fluxes (a-b). We defined 10 sub-regions where TWSs are highly correlated with PME in NATO3 (the cross correlation > 0.50) as high-correlated sub-regions (HSR), and defined other sub-regions as low-correlated sub-regions (LSR). (c) displays the spatial distribution of NATO1-4, LSRs, and HSRs respectively. (d-g) are the lags for maximum cross correlations between TWSs in HSRs (LSRs) and PMEs in NATOs1-4, respectively.

Supplementary Figure 10. Temporal variations in the normalized trend items of zonal sums of TWSs in sub-regions 1-10 (HSRs 1-10), and the normalized trend items of zonal sums of PME in the sub-regions 2 and 4 of North Atlantic Ocean (NATO2 and NATO4) during 2003-2017 by month (a-j). The Cross PR in the plot refers to the cross correlation coefficient.

Supplementary Figure 11. Pearson correlation between the normalized trend items of the regional sums of terrestrial water storages (TWSs) in the north (south) Xinjiang and normalized trend items of the regional sums of TWSs in HSR1-10 (a). (b-k) are the temporal variations in the normalized TWSs in north (south) Xinjiang and normalized TWSs in HSRs 1-10. The blue * indicates significant at 95% significance level.

6. Line 182-183. How you consider time lag in defining eastward and landward water vapor transport route.

Reply: Thank you for your kind review and professional suggestions. As for the landfalling drought originated from the NATO to Eurasia, the eastward and landward water vapor transport are the same water vapor transport route. As for the time lag in the transport route, here we applied the FLEXPART model to backward simulate the transports and water vapor variations in 500,000 particles with Xinjiang as the target region at 6-hourly time step based on the ERA-interim meteorological data ($0.75^\circ \times 0.75^\circ$) during 2003-2017. During the study period, we performed 180 backward simulations from day 1 to day 10 at 6-hourly time step for every month. Thus, for an eastward and landward water vapor transport trajectory, the longest time lag between the initial particle position and last particle position is 10 days in a simulation.

7. Figure 2 What the 'a,b,c...j' stand for....

Reply: Thank you for your kind review and professional suggestions. In Fig. 2, the 'a, b, c ..., j' refers to the temporal variations in TWSs in HSR1-10 and PME over the

NATO1 and NATO3. In Fig. 2g, we marked the spatial locations of the HSR1-10 with the symbols “(a)-(l)”.

Figure 2 Temporal covariations among the normalized trend items of the regional sums of terrestrial water storages (TWSs) in sub-regions 1-10 (HSRs 1-10) across mid-latitude (N30°-N60°) Eurasia and the normalized trend items of the regional sums precipitation-minus-evapotranspiration (PMEs) in NATO1 and NATO3 (a-e and h-l, respectively). Spatial patterns of the modified Mann-Kendall (MMK) trends of PMEs in the NATO and TWSs in North Africa, Europe and Asia (g). Cross Pearson’s correlation between PMEs in northern NATO and TWSs in HSRs 1-10. PME_NATO_MMK refer to the MMK trend in PMEs over NATO. TWS_MMK refers to the MMK trend in TWS across Eurasia and North Africa. The ocean currents in the plot were the long-term averaged ocean currents in NATO based on the OSCAR (Ocean Surface Current Analysis Real-time) dataset during 1993-2019. The moisture trajectories in subplot (g) were clustered using the k-means cluster method based on the locations of particles in the trajectories.

8. Line 310-312, Please update the descriptions.

Reply: Thank you for your kind review and professional suggestions. We have updated the descriptions in the revised manuscript by separately describing two seasonal paths. For your convenience, we have listed the specific revision as follows.

Former description:

Meanwhile, we identified two seasonal landfalling routes for the propagation of meteorological drying from LNATO along LNATO-North Africa (Europe)-Caspian Sea-Xinjiang-NCP in January-May (June-January).

Revised description:

Meanwhile, we identified two seasonal landfalling routes of meteorological drying from LNATO: (1) LNATO-North Africa-Caspian Sea-Xinjiang-NCP during January-May; (2) LNATO-North-Africa-Europe-Caspian Sea-Xinjiang-NCP during June-January.

9. Line 314, TWS decrease doesn't surely mean water scarcity.

Reply: Thank you for your kind review and professional suggestions. We have updated the descriptions in the revised manuscript. For your convenience, we have listed the specific revision as follows.

In summary, we demonstrated the mechanistic linkages between meteorological drying in LNATO (oceanic meteorological behaviors) and decline in TWS across MLE (continental water availability), clarifying and elucidating the key atmospheric drivers behind the decline in TWS across the MLE.

10. Low-latitude NATO is consistent with the decrease of TWS along the HSR1-10, while high- latitude NATO shows less or uncorrelated. Further explanations are needed for the reason.

Reply: Thank you for your kind review and professional suggestions. We did the maximum covariance analysis to evaluate the spatiotemporal covariance between PME over NATO and TWS across the mid-Latitude Europe, where HSR1-10 located. The following provides the detailed description of the results of the maximum covariance analysis.

Reply Fig. 9 indicated that the explained variances of the leading modes 1-4 are 28.72%, 13.09%, 9.29% and 7.12%, respectively (Supplementary Text 1). The temporal sequences of PME and TWS in the leading modes 1-4 are highly related with the correlation coefficients in the range of 0.86-0.94 ($p < 0.05$). It verified that the changing tendencies in PME and TWS were in significantly coherency (Reply Figs. 9a, d, g, j). Especially, in the first leading mode, the changing tendencies in PME over the low-latitude NATO were coherent with changing tendencies in TWS across the mid-latitude Eurasia (Reply Figs. 9a-c). The changing tendencies in PME over the high-latitude NATO were coherent with changing tendencies in TWS in the southern, eastern and northern Europe and most area of the western Russia (Reply

Figs. 9a-c). This spatial pattern agrees with the propagation routes of the landfalling impacts of deficits in PME over northern NATO from ocean to land during 2003-2017 in the Figure 3b in the main text (Reply Figs. 9a-c).

Reply Figure 9. (Supplementary Figure 14). The maximum covariance analysis (MCA) done between the TWS across the Eurasia and PME over the NATO. (a, d, g, j) are the temporal sequences for TWS and PME in the leading modes 1-4. (b, e, h, k) are the spatial modes of PME in the leading modes 1-4. (c, f, i, l) are the spatial modes of TWS in the leading modes 1-4.

Figure 3. Longitudinal profile of monthly-averaged mean precipitation-minus-evapotranspiration (PME) anomalies in the mid-latitude (N30°-N60°) Atlantic Ocean (NATO) and the monthly mean terrestrial water storage (TWSs) anomalies in mid-latitude (N30°-N60°) North Africa (NAF) and Eurasia from January to December (a). Spatial patterns of the sum of monthly mean PME anomalies (unit: mm) in NATO and the sum of negative monthly mean TWS anomalies (unit: mm) in lands during 2003-2017 (b). Propagation routes of the landfalling impacts of deficits in PME over northern NATO from ocean to land during 2003-2017 (c). The NXJ, SXJ, TM, HC, LP and NCP in subplot c refer to North Xinjiang, South Xinjiang, Tian Mountain, Hexi Corridor, Loess Plateau and North Chain Plain, respectively.

Besides, we also performed the attribution analysis between the TWS across the mid-latitude Eurasia and the total freshwater withdrawal (TFW) and PME over the NATO3 and NATO4 (Reply Fig. 1). Among the regions where the TWS were in increasing trends, the TWS were in highly coherency with the variation in the PME over the NATO4. These regions mostly distributed in the southern, eastern and northern Europe and most area of the western Russia, where is close to NATO4, i.e., Spain, France and Netherlands.

In contrast, among the regions where the TWS were in declining trends, Bosnia and Herzegovina (BosHerz for abbreviation in plot), Lebanon, Pakistan, Tunisia and Xinjiang (China) were the merely 5 regions featured by the TFW-dominated TWS (Reply Fig. 1e). Despite of the differences in the variations of TFW, the declines in TWS in these regions were all coherent with the decline in the PME over the NATO3 (Reply Fig. 1g-k). It should be noted that the variations in TWS in Bosnia and Herzegovina, Tunisia and Xinjiang (China) all sensitively responded to the abrupt change in PME over the NATO3 in 2015 (Reply Fig. 1g-k). Besides, in comparison with the regions where TWS was in increasing trend, the regions featured by declining TWS mainly distributed along the eastward propagation route of the landfalling PME-deficit originated from the low-latitude NATO. The temporal variation and spatial pattern both demonstrated that the PME deficit originated from the low-latitude NATO induced the widespread TWS declines across the mid-latitude Eurasia. Due to the anti-drought demands, featured by above 5 regions, TFW exacerbated the declines in TWS in the mid-latitude Eurasia.

Reply Figure 1 (Supplementary Figure 18). Attribution analysis for the variation in the TWS across the Eurasia on the region/country scale. (a-c) refer to the coefficients for the TFW, PME over the NATO3 and PME over the NATO4, respectively. (d) refers to the difference between the absolute value of the coefficient a and sum of the absolute value of the coefficient b and c. And the negative difference value indicates the variation in TWS is majorly influenced by the TFW (TFW-dominated TWS) while the positive difference value indicates the variation in TWS is majorly influence by the variation in PME over the NATO (PME-NATO-dominated TWS). (e-f) refer to the spatial patterns of the regions with the TFW-dominated TWS or PME-NATO-dominated TWS where the TWS were in declining (e) or increasing (f) trends. (g-o) refer to the temporal variations in TWS, TFW, PME-NATO3 and PME-NATO4 at regions with TFW-dominated TWS during 2003-2016.

11. Line 317 The findings indicate that atmospheric drivers have played broader TWS declines across MLE, however evidence is lack to show it is the key role.

Reply: Thank you for your kind review and professional suggestions. We did the attribution analysis for the variation in TWS across the Eurasia with consideration of

interactions between total water withdrawal (TFW) and variations in PME over NATO1 and NATO3. The detail introduction of the attribution analysis was included in the Supplementary Information and reads as included below. In the main text, we have included the key conclusions of the attribution analysis.

In the main text:

An attribution analysis indicated that TWS changes over 71% of the regions—where the total freshwater withdrawals (TFW) had negative impacts on TWS variations—were influenced mainly by PME variations over NATO, whereas TWS changes over 29% of these regions were impacted primarily by TFW (Supplementary Fig. 18). Besides, an evaluation of interactions between TFW and TWS suggested that variations in TWS across the MLE can affect water withdrawals at the regional scale, influencing regional economic development (Supplementary Fig. 19 and Supplementary Text 3).

Detailed introduction of the attribution analysis:

We have collected the annual total freshwater withdrawal (TFW) data at a national/region scale across the mid-latitude Eurasia during 2003-2016. These data are sourced from the Organization for Economic Co-operation and Development (OECD, <https://data.oecd.org/water/water-withdrawals.htm>) and the World Bank (<https://data.worldbank.org/>). The annual TFW data for the Xinjiang of China and the North China Plain were obtained from the China Data Insights (<https://cdi.cnki.net>). Here, the TFW was defined as the total freshwater withdrawals by domestic, agricultural and industrial activities.

We developed a linear regression model (Eqs. 1-3) evaluating relations between TWS (Terrestrial Water Storage), TFW and PME (precipitation-minus-evapotranspiration) over the NATO, quantifying fractional contributions of human activities and climate changes to variations in TWS over Eurasia. The absolute coefficients for each factor were used to evaluate attributions of each factor to variations in TWS. The linear regression model and related coefficients were introduced as follows:

$$TWS = a \times TFW + b \times PME_{NATO3} \text{ (declining TWS)} \quad (\text{Eq. 1})$$

$$TWS = a \times TFW + b \times PME_{NATO3} + c \quad (\text{Eq. 2})$$

$$\begin{aligned} &\times PME_{NATO4} \text{ (increasing TWS)} \\ \text{diff} = &abs(a) - (abs(b) + abs(c)) \quad (\text{Eq. 3}) \end{aligned}$$

where a, b and c are the coefficients for TFW and PME over NATO3 and PME over NATO4, respectively. For most regions across MLE where TWS had declining trends, the combined impacts from the variations in TFW and declines in PME over NATO3 were evaluated using Eq.1. For regions where TWS showed increasing trends and was spatially close to NATO4, the combined impacts from variations in TFW, decline in PME over NATO3, and increase in PME over NATO4 were evaluated using Eq. 2. Difference between the coefficients was estimated using Eq. 3. A negative diff indicated that TWS variation was dominated by the influence from TFW (TFW-dominated TWS), while a positive diff indicated that TWS variation was

dominated by the variations in PME over NATO (PME-NATO-dominated TWS). Since the impact of TFW on TWS variation is negative, the computation with $a > 0$ was not included in the attribution analysis. However, we identified an interesting phenomenon behind the positive a , which is discussed with details in Supplementary Text 3. Out of your convenience, the detailed discussion has been listed as follows.

Reply Figs. 1a-d indicated that TWS in 71% of the regions with negative a coefficient was influenced mainly by variations in PME over the NATO; while TWS in 29% of these regions was influenced mainly by TFW. Only 5 out of the regions with decreasing TWSs were featured by TFW-dominated TWS such as Bosnia and Herzegovina (BosHerz for abbreviation in plot), Lebanon, Pakistan, Tunisia and Xinjiang (China) (Reply Fig. 1e). Despite of the differences in variations of TFW, the decrease in TWS in these regions were in good line with the decrease PME over the NATO3 (Reply Figs. 1g-k). It should be emphasized that the variations in TWS in Bosnia and Herzegovina, Tunisia and Xinjiang (China) are highly sensitive to abrupt changes in PME over the NATO3 in 2015 (Reply Figs. 1g-k). Meanwhile, when compared to regions with increasing TWS, the regions with decreasing TWS mainly distributed along the eastward propagation route of the landfalling PME-deficit originated from the low-latitude NATO. The temporal variation and spatial pattern both demonstrated that the PME deficit originated from the low-latitude NATO induced widespread decrease in TWS across the mid-latitude Eurasia. For the sake of drought mitigation in abovementioned 5 regions, TFW can exacerbate the deficit of TWS in the mid-latitude Eurasia.

France, Netherlands, Spain and Sweden were mere 4 out of the regions with increasing TWSs which were featured by TFW-dominated TWS (Reply Figs. 1f). However, the declines in TFW in France, Netherlands and Spain did not significantly restore the TWS during 2003-2009 and the increase in TFW in Sweden did not significantly lead to the decline in the TWS during 2003-2009 (Reply Figs. 1l-o). Despite of the spatial heterogeneity of TFW over regions with increasing TWSs, the TWS over above-mentioned 4 regions was in good line with variation in PME over the NATO4 during 2003-2015. Besides, different from the regions with the declining TWS, the regions featured by the increasing TWS distributed mostly in the western, southern and northern Europe, where is spatially close to the NATO4 (Reply Figs. 1f). In summary, it demonstrated that the increase in PME over the NATO4 directly triggered increasing trends in TWS while decline in TFWs did not significantly impact the TWS over the western, southern and northern Europe during 2003-2016.

Reply Figure 1. (Supplementary Figure 18). Attribution analysis for TWS variation across the Eurasia at the regional/national scale. (a-c) refer to the coefficients for the TFW, PME over the NATO3 and PME over the NATO4, respectively. (d) refers to the difference between the absolute value of the coefficient a and sum of the absolute value of the coefficient b and c . The negative difference value indicates that TWS variation is influenced mainly by the TFW (TFW-dominated TWS); while the positive difference value indicates that TWS variation is influenced mainly by PME variation over the NATO (PME-NATO-dominated TWS). (e-f) refer to the spatial patterns of the regions with TFW-dominated TWS or PME-NATO-dominated TWS with decreasing (e) or increasing (f) TWS. (g-o) refer to the temporal variations in TWS, TFW, PME-NATO3 and PME-NATO4 at regions with TFW-dominated TWS during 2003-2016.

With exception of human impacts on water storage, it is interesting to find influence of variation in water storage on freshwater withdrawal behavior at regional scale, which is reflected by the positive coefficient a , implying synchronous changes between TFW and TWS (Reply Fig. 2). It violated the assumption that TFW has

negative impacts on TWS. Meanwhile, we evaluated relations between TWS and TFW and found that the correlation coefficient is > 0.3 over 65% of these regions (Reply Figs. 2), such as Belgium, Denmark, Greece, Poland, Israel, Slovenia, Georgia, Iraq, Kyrgyzstan, Kazakhstan, Moldova, Montenegro, Syria, Tajikistan and Ukraine. Moreover, variations in TWS were in agreement with variations in TFW and the increasing rates of GDP (gross domestic production) (Reply Figs. 2c-q). Of course, it makes no sense if we attempt to elucidate these phenomena from the perspective of the negative impacts of the local economic development and TFW on TWS. Instead, these phenomena demonstrated that variation in TWS could impact the freshwater withdrawal behavior at regional scale, and further influence the local economic development due to constraint of the freshwater withdrawals.

In summary, we compared impacts of TFW and PME over the NATO on TWS at regional scale, and evaluated interactions between TFW and TWS. The TWS across the mid-latitude Eurasia was dominated by PME over the NATO in 71% of the regions and was mainly influenced by TFW in 29% of the regions. Besides, we also demonstrated that variation in TWS could impact the freshwater withdrawal behavior at regional scale, and could further influence the local economic development.

Reply Figure 2. (Supplementary Figure 19). The impact of decline in TWS across the mid-latitude Europe on the freshwater withdrawal behavior and local economic development. (a) Spatial pattern of the modified Mann-Kendall trends of the TWS and relations between TWS and TFW at national/regional scale; (b) Statistic analysis of the relations between TWS and TFW at national/regional scale. Red filled circles marked by white cross show significant relations at 95%. (c-q) Temporal variations in TWS, TFW and growing magnitude of Gross Domestic Product (GDP_GR).

12. Too many abbreviations add the difficulty in understanding the manuscript

Reply: Thank you for your kind review and professional suggestions. You are right that too many abbreviations may add difficulty in understanding the manuscript. We used abbreviations to keep the text concise and meet the journal's length requirements.

While we appreciate your thoughts, we have opted to keep the abbreviations because, with the addition of new text during the revision, we couldn't afford to further lengthen the manuscript. We have, however, included a detailed table listing the main abbreviations in the Supplementary Information. We hope this address the issues to certain extent.

Supplementary Table 1 The collection of the abbreviation in the paper

Index	Abbreviation	Clarification
1	MLE	mid-latitude Eurasia
2	TWS	Terrestrial water storage
3	LNATO	low-latitude North Atlantic Ocean
4	PME	precipitation-minus-evapotranspiration
5	GRACE	Gravity Recovery and Climate Experiment
6	ENSO	El Niño-Southern Oscillation
7	NATO	North Atlantic Ocean
8	NATO1-4	sub-regions 1-4 of NATO
9	ERA5	ECMWF Reanalysis 5th generation
10	CMIP6	Coupled Model Intercomparison Project Phase 6
11	SSP245	Shared Socioeconomic Pathway 2-4.5
12	SSP585	Shared Socioeconomic Pathway 5-8.5
13	FLEXPART	Lagrangian transport and dispersion model
14	XJ	Xinjiang
15	AS	AS, Asia excluding Xinjiang
16	EU	Europe including Russia
17	TCR	total contribution rate (TCR)
18	GPCC	Global Precipitation Climatology Centre
19	CR	Contribution rates
20	EU1-3	sub-regions 1-3 of EU
21	AS1-2	sub-regions 1-2 of EU
22	NATO1-4	sub-regions 1-4 of NATO
23	HSR	sub-regions where TWSs were highly correlated with PME in NATO3 (the cross correlation > 0.50) as high-correlated sub-regions
24	LSR	sub-regions where TWSs were lowly correlated with PME in NATO3 (the cross correlation < 0.50 as low-correlated sub-regions (LSR))
25	MMK	Modified Mann-Kendall trend
26	SST	Sea Surface Temperature
27	NCP	Northern China Plain
28	CS	Caspian Sea
29	NAF	North Africa
30	NAM or NA	North America
31	MS	Mediterranean Sea

32	AO	Arctic Ocean
33	BS	Black Sea
34	IO	Indian Ocean
35	RS	Red Sea
36	PO	Pacific Ocean
37	TFW	Total freshwater withdrawal

Reviewers' Comments:

Reviewer #1:

Remarks to the Author:

I congratulate authors for providing a detailed response for each of my comments. The manuscript has greatly improved after the revision. I have no further comments and the manuscript can be accepted.

Reviewer #2:

Remarks to the Author:

All my concerns have been well addresses, therefore I suggest the acceptance of the manuscript.

Reviewer #1 (Remarks to the Author):

I congratulate authors for providing a detailed response for each of my comments. The manuscript has greatly improved after the revision. I have no further comments and the manuscript can be accepted.

Answer: Thanks for your detailed review and comments, which does improve our manuscript quality to a great extent. We really appreciate your sincere suggestion.

Reviewer #2 (Remarks to the Author):

All my concerns have been well addresses, therefore I suggest the acceptance of the manuscript.

Answer: Thanks for your detailed review and comments, which does improve our manuscript quality to a great extent. We really appreciate your sincere suggestion.